# Beyond Value Functions: Single-Loop Bilevel Optimization under Flatness Conditions

**Liuyuan Jiang**[*,⋄]**, Quan Xiao**[†,⋄]**, Lisha Chen**[*]**, Tianyi Chen**[†,⋄]
⋄Rensselaer Polytechnic Institute, Troy, NY
*University of Rochester, Rochester, NY
†Cornell Tech, Cornell University, New York, NY
ljiang24@ur.rochester.edu, qx232@cornell.edu
lisha.chen@rochester.edu, tianyi.chen@cornell.edu [*]

## Abstract

Bilevel optimization, a hierarchical optimization paradigm, has gained significant attention in a wide range of practical applications, notably in the fine-tuning of generative models. However, due to the nested problem structure, most existing algorithms require either the Hessian vector calculation or the nested loop updates, which are computationally inefficient in large language model (LLM) fine-tuning. In this paper, building upon the fully first-order penalty-based approach, we propose an efficient value function-free (PBGD-Free) algorithm that eliminates the loop of solving the lower-level problem and admits fully single-loop updates. Inspired by the landscape analysis of representation learning-based LLM fine-tuning problem, we propose a relaxed flatness condition for the upper-level function and prove the convergence of the proposed value-function-free algorithm. We test the performance of the proposed algorithm in various applications and demonstrate its superior computational efficiency over the state-of-the-art bilevel methods.

## 1 Introduction

Bi-level optimization (BLO) has gained significant attention for its powerful modeling capabilities in hierarchical learning across a wide range of real-world applications, such as distributed learning [69, 28], meta learning [27, 12, 27], model pruning [111, 96], reinforcement learning [108, 84, 79, 85], continual learning [8, 33], fine-tuning large language models (LLMs) [76, 68, 64, 56, 107] and diffusion models [93, 17, 61]. In this paper, we consider the BLO problem with $f : \mathbb{R}^{d_x} \times \mathbb{R}^{d_y} \to \mathbb{R}$ and $g : \mathbb{R}^{d_x} \times \mathbb{R}^{d_y} \to \mathbb{R}$ being the upper-level (UL) and lower-level (LL) objectives that are continuously differentiable but not necessarily convex. Since the LL problem may contain multiple solutions in $S_g^*(x)$, we consider the *optimistic* BLO formulation which selects the one $y$ that minimizes the UL objective, given by

$$\min_{x,y} \ f(x,y) \ \text{ s.t. } \ y \in S_g^*(x) := \arg\min_y g(x,y). \tag{1}$$

In large-scale machine learning problems, efficiency is given a higher priority [110] and it is critical to use gradient-based approaches to solve the above problem. One can perform a direct gradient descent (GD) on the hyper-objective $\phi(x) := \min_{y \in S_g^*(x)} f(x,y)$. A popular GD-based method is the implicit gradient descent (IGD) method with second-order Hessian evaluation [29, 34, 39, 15, 42, 47, 82]. However, evaluating Hessian or its inverse in IGD is costly. To reduce the computational burden, especially in large-scale problems, first-order gradient-based methods, including the penalty-based BLO methods [105, 50, 77, 44, 45, 40, 57], have been developed. For example, penalizing the LL

---

[*]The work was supported by the National Science Foundation Projects 2401297, 2532349 and 2532653, and by the Cisco Research Award.

39th Conference on Neural Information Processing Systems (NeurIPS 2025).

objective optimality gap into the UL via a large penalty constant $\gamma$ has been proposed [105, 78, 45, 58], yielding the following objective

$$\min_x F_\gamma(x) := \min_y \tilde{F}_\gamma(x,y) := f(x,y) + \gamma\big(g(x,y) - \min_z g(x,z)\big). \tag{2}$$

Under a proper curvature assumption for the LL problem, the penalty reformulation proves to be differentiable and smooth [78, 45, 40, 11], which enables the design of penalty-based gradient descent algorithms (PBGD) [78, 44, 45, 10]. Furthermore, the function value gap $|F_\gamma(x) - \phi(x)| = \mathcal{O}(\gamma^{-1})$ ensures the solution to the reformulated problem is an approximate solution to the original problem. The reformulation in (2) provides two choices of algorithm update: jointly updating $x$ and $y$ to minimize $\tilde{F}_\gamma(x,y)$ [105, 78], or alternatively optimizing $y$ then updating $x$ to minimize the hyper objective $F_\gamma(x)$ [45, 11]. Each has its pros and cons. For example, joint update eliminates the inner loop of $y$ so that it has low per-iteration cost, but high smoothness constant of $\tilde{F}_\gamma(x,y)$, which increases with $\gamma$, making the convergence rate suboptimal [78, 105]. In contrast, the smoothness constant of $F_\gamma(x)$ remains $\mathcal{O}(1)$ since the value function gap $\gamma(g(x,y) - \min_z g(x,z))$ remains in $\mathcal{O}(1)$ when $y$ minimizes $\tilde{F}(x,y)$, but estimating $\nabla F_\gamma(x)$ often requires running inner loops to obtain $y_g^*(x) \in S_g^*(x)$ and $y_\gamma^*(x) \in S_\gamma^*(x) := \operatorname{argmin}_y \tilde{F}_\gamma(x,y)$ [11]. Then a natural question is:

**(Q1)** *Can we develop an efficient algorithm that combines the best of both worlds?*

The idea is to update $x$ by $\nabla_x f(x, y_\gamma^*(x))$ and skip the inner loop estimation for $y_g^*(x)$, which we term as PBGD Free of value function evaluation (PBGD-Free). To be more specific, we illustrate the updates for standard PBGD, its variants, and PBGD-Free in Figure 1.

**Positive empirical observations on Q1:** *PBGD-Free largely reduces computation and memory cost while preserving the accuracy in LLM parameter efficient fine-tuning (PEFT) [72, 2].* See Figure 2 and experiments in Section 4. We prioritize supervised fine-tuning (SFT) loss at the LL to ensure a capable base LLM model, while we keep direct preference optimization (DPO) loss [70] in the UL to keep alignment with human preferences:

$$\min_{x,y} f_{\text{DPO}}(x,y; \mathcal{D}_{\text{DPO}}) \tag{3}$$

s.t. $y \in \arg\min_y g_{\text{SFT}}(x,y; \mathcal{D}_{\text{SFT}})$,

where $x$ is a pretrained LLM model, and $y$ is an easy-to-fine-tune head. This design is aligned with both theoretical and practical needs in LLM deployment, as detailed in Section 4.

**Negative theoretical observations on Q1:** *There are some counterexamples where PBGD-Free does not converge.* **c1)** When the UL objective solely depends on the LL variable $f(x,y) = f(y)$, as in data hyper-

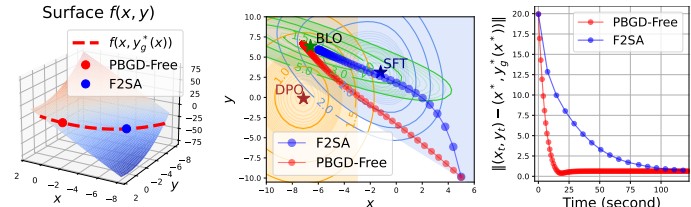

Figure 1: Update schemes for V-PBGD, F²SA and PBGD-Free. V-PBGD [78] (**top**) and F²SA [45] (**middle**) refine the LL variable over multiple steps before updating $x_t$ via $\nabla_x \tilde{F}_\gamma(x_t, y_t)$ for V-PBGD or $\nabla_x \tilde{F}_\gamma(x_t, y_t)$ for F²SA while PBGD-Free (**bottom**) applies a 1-step inner update to find a more efficient yet potentially less accurate $\nabla_x f(x_t, y_{t+1}) \approx \nabla F_\gamma(x_t)$.

Figure 2: **An Illustration to show PBGD-Free does not work in Example 1, but works well in PEFT.** The **left** plot shows the $f(x,y)$ and $f(x, y_g^*(x))$ in Example 1, with red and blue dots as the converged points using PBGD-Free and F²SA method. The **middle** plot shows the trajectory of updates in PEFT. The orange, cyan, and green contours are the landscapes of $f_{\text{DPO}}(x,y)$, $g_{\text{SFT}}(x,y)$, and $\tilde{F}_\gamma(x,y)$, respectively. The **right** plot presents the convergence vs. time in PEFT, showing faster convergence of PBGD-Free. (See Appendix C.1 for details.)

cleaning [75, 34, 76] and meta-learning [27, 12, 27], the LL penalty gradient term contains all the gradient information about the UL variable $x$, so it cannot be omitted; and, **c2)** When $f(x,y)$ jointly depends on both variables, omission of the penalty gradient term can lead to a different update direction; see more details in Example 1 and Proposition 2.

| Property | PBGD-Free | V-PBGD | BOME | F$^2$SA-MA | F$^2$SA | BVFSM |
|---|---|---|---|---|---|---|
| $f(x,\cdot)$ | Flat | Lip | Lip & B | Lip | Lip | Diff |
| $g(x,\cdot)$ | PL | PL | PL & B | PL | PL | Diff |
| $f(x,y) + \gamma g(x,y)$ | PL | / | / | PL | PL | / |
| Single-loop | ✓ | ✗ | ✗ | ✓ | ✗ | ✗ |
| Memory cost | $d_x + d_y$ | $d_x + 2d_y$ | $3d_x + 4d_y$ | $d_x + 5d_y$ | $d_x + 2d_y$ | $d_x + 2d_y$ |
| Complexity | $\mathcal{O}(\epsilon^{-1})$ | $\tilde{\mathcal{O}}(\epsilon^{-1.5})$ | $\tilde{\mathcal{O}}(\epsilon^{-4})$ | $\mathcal{O}(\epsilon^{-1.5})$ | $\tilde{\mathcal{O}}(\epsilon^{-1})$ | Asym |

Table 1: Comparison of the proposed method (PBGD-Free) with the existing first-order approaches for BLO with nonconvex LL problem (PBGD [78], BOME [105], F$^2$SA-MA [45] (with momentum assistance), F$^2$SA [11] and BVFSM [51]) in deterministic setting. The notation $\tilde{\mathcal{O}}$ hides dependency on $\log(\epsilon^{-1})$ terms. 'Flat', 'Lip', 'B', 'Diff', and 'Asym' stand for 'flatness condition' in Def. 1, 'Lipschitz continuous', 'bounded', 'differentiable', and 'asymptotic convergence'.

**Example 1.** *For the BLO problem in* (1) *with* $f(x,y) = x^2 + 10y$ *and* $g(x,y) = (y - x + 1)^2$, *the gradients* $\langle \nabla F_\gamma(x), \nabla_x f(x, y^*_\gamma(x)) \rangle < 0$ *exhibit* **opposite directions** *for* $x \in (-5, 0)$. *As a result,* $\nabla F_\gamma(x) = 2x + 10$ *converges to* $x = -5$ *while* $\nabla_x f(x, y^*_\gamma(x)) = 2x$ *converges to* $x = 0$.

These findings leave it unclear when PBGD-Free can be applied without sacrificing accuracy. In this paper, we focus on the case with UL joint dependency, where the objective $f(x, y)$ depends intrinsically on both $x$ and $y$ (i.e., cannot be simplified to $f(y)$). We explore the following question:

**(Q2)** *Can we identify sufficient conditions under which the PBGD-Free algorithm is guaranteed to converge to the stationary solution of the original problem?*

We give an affirmative answer to the above question. Specifically, **our key contributions** are summarized as follows, and the broader impact is discussed in Appendix D.

- **C1)** We propose PBGD-Free, a computationally efficient fully-single-loop, value-function-free, first-order algorithm. See a detailed comparison with other algorithms in Table 1. Specifically, compared to V-PBGD, it reduces the memory cost from $O(d_x + 2d_y)$ to $O(d_x + d_y)$, and the per-iteration computational complexity cost from $O(K)$ to $O(1)$, where $K$ is the number of inner iterations. Furthermore, we show that empirically, it works in large-scale problems such as PEFT (3). But theoretically, under a *Lipschitz condition* on the UL objective, PBGD-Free only converges to an $\Theta(1)$-neighborhood of a stationary point.

- **C2)** We then introduce a Hölder-alike condition to describe the flatness of $f(x, \cdot)$ (see Definition 1), which relaxes the standard $l_{f,0}$-Lipschitz continuity assumption when $l_{f,0}$ is small. This condition allows us to establish an improved complexity of the PBGD-Free algorithm in $\mathcal{O}(\epsilon^{-1})$ (Theorem 3) to a necessary stationary condition of the original problem.

- **C3)** We validate our methods through applications to LLM with PEFT and bilevel low-rank adaptation. Across all experiments, PBGD-Free demonstrates much better efficiency and comparable or better accuracy than the state-of-the-art baselines. See Section 4.

## 1.1 Prior art

**Second-order BLO methods.** The convergence for IGD-based BLO approaches was firstly established for the unconstrained strongly-convex LL problem [29], with later literature focused on improving finite time convergence rate [29, 34, 38, 14, 15, 42, 47, 82, 16, 97, 39]. Another branch of methods is based on iterative differentiation (ITD) methods [60, 26, 63, 75, 7], but they generally lack finite-time guarantee under stochastic setting [32, 37]. However, convergence analysis for both ITD and IGD methods mentioned above is limited to the setting where the LL problem is strongly convex over $y$. This assumption does not align with large-scale machine learning applications, where the LL objective represents the loss of a neural network and is inherently nonconvex [86, 41]. Recent studies have generalized the IGD and ITD methods to address BLO with convex [81, 52] or even nonconvex LL problem [92, 3, 49, 50, 66, 98]. Nevertheless, both ITD and IGD require the computation of second-order information, making them inefficient for large-scale machine learning problems.

**First-order BLO methods.** Fully first-order bilevel methods based on equilibrium backpropagation [74, 74, 113] and penalty reformulation [78, 44, 45, 105, 11, 103] have become increasingly popular due to their computational efficiency and the ability to handle nonconvex LL problems. Later, penalty

approaches have been generalized to address BLO with constrained LL problem [100, 40] and distributed learning settings [99, 87]. However, the iteration complexity of fully first-order approaches remains suboptimal, exhibiting a logarithmic dependency due to the inner-loop overhead. To reduce the cost in inner loops for both $y_g^*(x)$ and $y_\gamma^*(x)$, PBGD [78] eliminated the inner loop for $y_\gamma^*(x)$ by jointly optimizing $x$ and $y_\gamma$ in (2), F$^2$SA [45] managed to be fully single-loop using momentum and warm-start techniques. However, both methods incur a suboptimal iteration complexity of $\mathcal{O}(\epsilon^{-1.5})$. [11] further improved iteration complexity of double-loop version of F$^2$SA by exploiting the fact that $F_\gamma(x)$ is $\mathcal{O}(1)$-Lipschitz smooth, but its inner loop leads to a high per-iteration computational cost and suboptimal convergence rate as $\mathcal{O}(\epsilon^{-1}\log(\epsilon^{-1}))$.

**Landscape-aware optimization.** Landscape-aware optimization leverages structural properties of objective functions into algorithm design to accelerate the convergence or improve the generalization. Newton-type methods, which use second-order curvature information to rescale gradients, have been utilized in BLO [23, 71, 21] for efficient Hessian-vector calculation in IGD-based BLO methods. Sharpness-aware minimization [25], which seeks solutions robust to local perturbations and promotes convergence to flat minima, has also been incorporated into BLO [1] for improved generalization. Other landscape conditions in single-level optimization, such as relaxed smoothness [109, 46] and Hessian spectrum [112, 30], are key to explaining the theoretical benefits of empirically effective algorithms like gradient clipping and Adam [43]. However, most existing works focus on second-order BLO algorithms, and none have explored BLO tailored landscape conditions.

## 2 Value Function Free Algorithm for BLO Problems

In this section, we introduce the value function free algorithm for bilevel problems and show that it does not always converge under the traditional Lipschitz condition.

### 2.1 Preliminary: the Lipschitzness condition and the penalty-based reformulation

We begin by introducing the standard Lipschitz condition on the UL objective $f(x, y)$, which is common in BLO analysis.

**Assumption 1.** *Assume that for all $x$, the UL objective $f(x, \cdot)$ is $l_{f,0}$-Lipschitz in $y$ at $y_g^*(x)$ with some $l_{f,0} > 0$, i.e.,*
$$|f(x, y_g^*(x)) - f(x, y)| \leq l_{f,0}\|y - y_g^*(x)\|. \tag{4}$$

For differentiable $f$, Assumption 1 implies $\|\nabla_y f(x, y_g^*(x))\| \leq l_{f,0}$. This assumption is crucial for the key results in BLO literature; e.g., [78, 45, 11, 15, 39, 34, 38, 105]. Together with the following standard assumption, it enables a good approximation of $F_\gamma(x)$ to the original problem.

**Assumption 2.** *Suppose that i) $f$ and $g$ are respectively $l_{f,1}$ and $l_{g,1}$-smooth; ii) $\nabla^2 f$ and $\nabla^2 g$ are respectively $l_{f,2}$ and $l_{g,2}$-Lipschitz continuous; and, iii) there exists a finite $\gamma^* > 0$ such that $cf(x, y) + g(x, y)$ is $\mu$-Polyak-Łojasiewicz (PL) in $y$ for all $c \in [0, 1/\gamma^*]$ for some $\mu > 0$.*

We provide the definition of smoothness and PL in Appendix A. Here, the smoothness condition is standard [29, 34, 44, 38, 15, 40]. The Hessian Lipschitzness helps establish the smoothness of $F_\gamma(x)$ with constant nonincreasing with $\gamma$ and is also conventional [45, 11, 19]. PL condition is weaker than the strong convexity assumption [15, 34, 29, 38, 19] and is conventional in the first-order BLO literature [45, 78, 11]. Under these, the penalty objective is differentiable [78, 45, 11, 63] and
$$\nabla F_\gamma(x) = \nabla_x f(x, y_\gamma^*(x)) + \gamma \left( \nabla_x g(x, y_\gamma^*(x)) - \nabla_x g(x, y_g^*(x)) \right) \tag{5}$$
with $\forall y_g^*(x) \in S_g^*(x)$ and $\forall y_\gamma^*(x) \in S_\gamma^*(x) := \arg\min_y F_\gamma(x, y)$. Moreover, the following proposition shows that $F_\gamma(x)$ is a good approximation of the original objective $\phi(x)$.

**Proposition 1** (Approximation error [78, 45]). *Under Assumptions 1–2, for any $x$, we have*
$$\|F_\gamma(x) - \phi(x)\| \leq \mathcal{O}(l_{f,0}^2\mu^{-1}\gamma^{-1}), \quad and \tag{6}$$
$$\|\nabla\phi(x) - \nabla F_\gamma(x)\| = \mathcal{O}(\|y_g^*(x) - y_\gamma^*(x)\|) \leq \mathcal{O}(l_{f,0}\mu^{-1}\gamma^{-1}).$$
*for some $y_g^*(x)$, $y_\gamma^*(x)$ defined in (5). Moreover, the bound for $\|y_g^*(x) - y_\gamma^*(x)\|$ is tight.*

Therefore, the PBGD type of algorithms [11, 44, 45, 105] proceed by approximating $y_t^\gamma \approx y_\gamma^*(x_t)$ and $y_t^g \approx y_g^*(x_t)$ via gradient descent and updating $x$ via
$$x_{t+1} = x_t - \eta g_t \quad \text{where} \quad g_t = \nabla_x f(x, y_t^\gamma) + \gamma \left( \nabla_x g(x, y_t^\gamma) - \nabla_x g(x, y_t^g) \right). \tag{7}$$

---

**Algorithm 1** PBGD Free from value function (PBGD-Free) algorithm

---

1: **Inputs:** initial point $x_0, y_0$; step sizes $\eta, \eta^\gamma$; counters $T, K$      $\triangleright$ $K = 1$ is a common choice
2: **for** $t = 0, 1, \ldots, T - 1$ **do**
3:      **for** $k = 0, 1, \ldots, K - 1$ **do**
4:          $y_{t,k+1}^\gamma = y_{t,k}^\gamma - \eta^\gamma\big(\gamma^{-1}\nabla_y f(x_t, y_{t,k}^\gamma) + \nabla_y g(x_t, y_{t,k}^\gamma)\big)$      $\triangleright$ set $y_{t,0}^\gamma = y_{t-1}^\gamma$
5:      **end for**
6:      $x_{t+1} = x_t - \eta g_t$, where $g_t = \nabla_x f(x_t, y_t^\gamma)$      $\triangleright$ set $y_t^\gamma = y_{t,K}^\gamma$
7: **end for**
8: **Outputs:** $(x_T, y_T^\gamma)$

---

## 2.2 Negative theoretical results of the PBGD-Free under Lipschitz condition

Although PBGD-type algorithms can achieve the state-of-the-art complexity $\mathcal{O}(\epsilon^{-1}\log(\epsilon^{-1}))$ in [11], their reliance on two inner loops can become computationally expensive for large-scale problems. While the overhead is manageable in small-scale settings, it may pose practical challenges as the model size grows. Nevertheless, empirical evidence in Figure 2 and real-world applications in Section 4 illustrate that it sometimes gives satisfactory results even if it directly updates $x_{t+1}$ and $y_{t,k}^\gamma$ by

$$x_{t+1} = x_t - \eta\nabla_x f(x_t, y_t^\gamma) \quad \text{and} \quad y_{t,k+1}^\gamma = y_{t,k}^\gamma - \eta^\gamma(\gamma^{-1}\nabla_y f(x_t, y_{t,k}^\gamma) + \nabla_y g(x_t, y_{t,k}^\gamma)) \quad (8)$$

which we name as PBGD-Free algorithm and is summarized in Algorithm 1.

Although PBGD-Free is computationally efficient by eliminating the inner loop estimates of $y_g^*(x)$, the removal of the value function part $b(x_t) := \gamma(\nabla_x g(x, y_\gamma^*(x)) - \nabla_x g(x, y_g^*(x)))$ in PBGD-Free introduces a non-negligible bias shown in Example 1. To see this, by Taylor's expansion, the omitted value function part $b(x_t)$ is in the order of

$$\|b(x_t)\| = \gamma\|\nabla_{xy}g(x, y_g^*(x))(y_\gamma^*(x) - y_g^*(x))\| + \mathcal{O}(\gamma\|y_g^*(x) - y_\gamma^*(x)\|^2). \quad (9)$$

Here, the second term $\mathcal{O}(\gamma\|y_g^*(x) - y_\gamma^*(x)\|^2) = \mathcal{O}(l_{f,0}^2\gamma^{-1})$ can be small enough with enlarging $\gamma$ following Proposition 1. For general settings where $\nabla_{xy}g(x, y_g^*(x)) \neq 0$, due to the first term and according to Proposition 1, the bias in (9) is tight as $\Omega(1)$. Therefore, in the general case where $\nabla_{xy}g(x, y_g^*(x)) \neq 0$, the PBGD-Free algorithm only drives the iterates to a neighborhood of the stationary point, which we will formally quantify as follows.

**Proposition 2** (Lower bound on asymptotic error)**.** *Under Assumptions 1 and 2, there exists a BLO problem where the iterates generated by PBGD-Free (Algorithm 1) converge to a neighborhood of a stationary point with a non-vanishing residual even when choosing step sizes appropriately, i.e.,*

$$\lim_{T\to\infty}\frac{1}{T}\sum_{t=0}^{T-1}\|\nabla F_\gamma(x_t)\|^2 = \lim_{T\to\infty}\Theta\left(\frac{1}{T}\sum_{t=0}^{T-1}\|\nabla_y f(x_t, y_g^*(x_t))\|^2\right) = \Theta\left(l_{f,0}^2\right). \quad (10)$$

The proof of Proposition 2 is available at Appendix A.1. Proposition 2 illustrates that PBGD-Free converges to the $\epsilon$ stationary point only when the bound for $\|\nabla_y f(x, y_g^*(x_t))\|$ (a.k.a $\ell_{f,0}$) is small. However, this is difficult to guarantee even in scenarios where PBGD-Free is effective, such as in representation learning based PEFT (3). This motivates us to explore a weaker condition than the small Lipschitz assumption on $f(x, \cdot)$, one that is more likely to hold in practice.

# 3 Theoretical Analysis under the $(\delta, \alpha)$-Flatness Condition

In this section, we will introduce a new relaxed condition to replace the widely used Lipschitz condition of the UL objective, discuss its use cases, and establish the convergence rate of the PBGD-Free algorithm under this condition.

## 3.1 A relaxed condition: UL $(\delta, \alpha)$-flatness and its validation on PEFT problem (3)

We first introduce a relaxed condition that is less restrictive, and therefore more general, than the conventional uniform Lipschitz assumption on $f(x, \cdot)$.

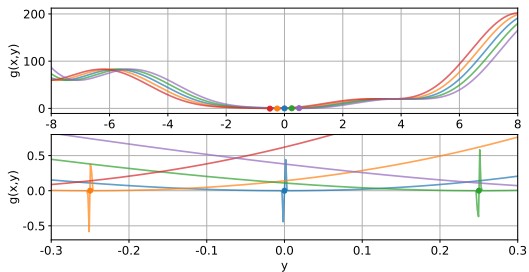

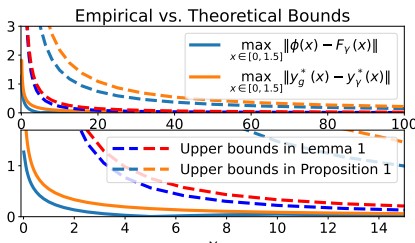

Figure 3: Visualization of $f(x,y)$ in Example 2 with details deferred to Appendix B.2. Colored curves represent $f(x,\cdot)$ for different $x$; dots show $(y_g^*(x), f(x, y_g^*(x)))$. The **upper** plot shows $f(x,\cdot)$ on a larger scale, and the **lower** one illustrates the fluctuation around $y_g^*(x)$.

Figure 4: Empirical bounds for $\|\phi(x) - F_\gamma(x)\|$ and $\|y_\gamma^*(x) - y_g^*(x)\|$ versus theoretical upper bounds in Proposition 1 and Lemma 1 for the illustration of representation learning PEFT (3). The lower plot shows a smaller scale.

**Definition 1** (($\delta, \alpha$)-flatness). *Function $f(x, \cdot) : \mathcal{Y} \to \mathbb{R}$ is called $(\delta, \alpha)$-flat with modulus $c \geq 0$ at $y_g^*(x) \in S_g^*(x)$ with $\delta \geq 0, \alpha \geq 1$ if $|f(x, y_g^*(x)) - f(x,y)| \leq c\|y_g^*(x) - y\|^\alpha + \delta$ holds for all $y$.*

When $\delta = 0$ and $y_g^*(x)$ is replaced by an arbitrary $y'$, Definition 1 reduces to the standard Hölder condition. Under Assumption 1, the function $f(x, \cdot)$ satisfies $(0, 1)$-flatness with modulus $c = l_{f,0}$. However, setting $\delta = 0$ naturally imposes the constraint $\alpha \leq 1$ whenever $\nabla_y f(x, y_g^*(x)) \neq 0$. Unless otherwise specified, we assume $\delta > 0$ and $\alpha > 1$ when referring to flatness in the following.

We then discuss the relations of Lipschitz condition in Assumption 1 and the new flatness condition in Definition 1 through several observations.

**Observation 1.** *Under the $l_{f,1}$-smoothness condition of $f(x, \cdot)$, if $\|\nabla_y f(x, y_g^*(x))\| = \delta^{\frac{1}{\alpha}}$, then $f(x, \cdot)$ is $(\delta, \alpha)$-flat with some modulus $0 \leq c \leq \mathcal{O}(l_{f,1})$.*

The proof of Observation 1 is provided in Appendix B.1. It demonstrates that assuming small $\|\nabla_y f(x, y_g^*(x))\|$ is stronger than assuming flatness since $\ell_{f,0} = \delta^{\frac{1}{\alpha}} > \delta$ when $\alpha > 1$. Below, we will show that the flatness condition automatically holds near the LL optimal solution $y_g^*(x)$.

**Observation 2.** *Under the smoothness condition of $f$, the $(\delta, \alpha)$-flatness condition holds automatically for all $y \in \{y : |f(x,y) - f(x, y_g^*(x))| \leq \delta\}$.*

Since $f$ is continuous and smooth, this observation implies that the flatness condition permits abrupt, unstable changes in the $\mathcal{O}(\delta)$-neighborhood of $y_g^*(x)$. This demonstrates that the flatness condition is relatively mild and further confirms that it is strictly weaker than the small Lipschitz condition, which explicitly requires $\|\nabla_y f(x, y_g^*(x))\|$ to be small. Figure 3 visualize an example that is $(3e^{-3}, 1.1)$-flat with modulus $c = 5$ at $y_g^*(x)$, but it exhibits a sharp change leading to a large Lipschitz continuity constant $\nabla_y f(x, y_g^*(x)) = 1000$. The details of Figure 3 are deferred to Appendix B.2.

Owing to the Hölder-alike condition, the following observation shows that outside of the $\mathcal{O}(\delta)$ neighborhood, the curvature of the flatness condition is also milder than the Lipschitz condition.

**Observation 3.** *Under $(\delta, \alpha)$-flatness, the growth rate of $f(x, \cdot)$ outside the $\mathcal{O}(\delta)$ neighborhood is*

$$\frac{|f(x,y) - f(x, y_g^*(x))|}{\|y - y_g^*(x)\|} \leq \begin{cases} \mathcal{O}(1), & \text{if } \mathcal{O}(\delta) \leq \|y - y_g^*(x)\| \leq \mathcal{O}(1), \\ \mathcal{O}\left(\|y - y_g^*(x)\|^{\alpha-1}\right), & \text{if } \|y - y_g^*(x)\| > \mathcal{O}(1). \end{cases} \quad (11)$$

This is obtained by dividing both sides of the flatness inequality by $\|y_g^*(x) - y\|$. For small $\|y_g^*(x) - y\|$, the second term dominates and leads to a $\mathcal{O}(1)$ bound, which is the same as the Lipschitz condition. However, for large $\|y_g^*(x) - y\|$, since $\alpha > 1$, the bound $\mathcal{O}(\|y_g^*(x) - y\|^{\alpha-1})$ can be larger than $\mathcal{O}(1)$. This observation further demonstrates that the flatness condition relaxes the Lipschitzness of $f(x, \cdot)$ in Assumption 1. Specifically, while Lipschitz continuity would require a uniform bound on the gradient, flatness allows for a higher growth rate of $\mathcal{O}(\|y - y_g^*(x)\|^{\alpha-1})$. For UL objective $f(x, \cdot)$ with fixed $x$, given a *pre-determined* $\alpha$ and modulus $c$, the $\delta$ constant for flatness condition in Definition 1 can be calculated via

$$\delta(x) := \max\{0, |f(x, y_g^*(x)) - f(x, y_\gamma^*(x))| - c\|y_g^*(x) - y_\gamma^*(x)\|^\alpha\}. \quad (12)$$

When $\|y_\gamma^*(x) - y_g^*(x)\| > \mathcal{O}(1)$, the last term in (12) dominates and $\delta(x)$ can effectively be 0. Therefore, together with Observation 2, the flatness condition with small $\delta$ not only encompasses a broader function class than small Lipschitz continuous functions, but is easier to hold in practice. For example, modern loss functions used in deep learning, such as cross-entropy, squared error, or exponential losses, are nonlinear and locally curved. Around $y_g^*(x)$, we can write $f(x, y') \approx f(x, y_g^*(x)) + c\|y' - y_g^*(x)\|^\alpha$ for some $\alpha > 1$ and constant $c > 0$. In such cases, the additive term in (12) vanishes and $\delta(x)$ is effectively zero. This implies that the flatness condition can hold even when no Lipschitz bound on $f(x, \cdot)$ is available, particularly for locally curved objectives. We next illustrate this behavior concretely through a parameter-efficient fine-tuning (PEFT) problem in representation learning.

### 3.2 The flatness of the representation learning PEFT problem.

In our PEFT framework in (3), the model, which can be any structure (e.g. CNN), is parameterized with $(x, y)$ by $\pi_{x,y}(r|z) := \mathrm{softmax}(\mathrm{model}_{x,y}(z))_r$. It gives the model's predicted probability for response $r$ given input question $z$. The DPO loss [70] over preference data $\mathcal{D}_{\mathrm{DPO}}$, compares outputs $\pi_{x,y}$ against a reference $\pi_{\mathrm{ref}}$ via

$$f_{\mathrm{DPO}}(x, y) := -\frac{1}{|\mathcal{D}_{\mathrm{DPO}}|} \sum_{(z, r_w, r_\ell) \in \mathcal{D}_{\mathrm{DPO}}} \log\left(\sigma\left(q_\beta(x, y; z, r_w, r_\ell)\right)\right), \quad (13)$$

where $q_\beta(x, y; z, r_w, r_\ell) := \beta \log \frac{\pi_{x,y}(r_w|z)}{\pi_{\mathrm{ref}}(r_w|z)} - \beta \log \frac{\pi_{x,y}(r_\ell|z)}{\pi_{\mathrm{ref}}(r_\ell|z)}$, $r_w$ and $r_\ell$ are the preferred and rejected responses to input $z$. The SFT loss operates on supervised dataset $\mathcal{D}_{\mathrm{SFT}}$ through

$$g_{\mathrm{SFT}}(x, y) := -\frac{1}{|\mathcal{D}_{\mathrm{SFT}}|} \sum_{(z, r_{\mathrm{SFT}}) \in \mathcal{D}_{\mathrm{SFT}}} \log\left(\pi_{x,y}(r_{\mathrm{SFT}}|z)\right). \quad (14)$$

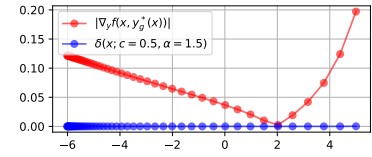

Both objectives are differentiable with the following gradients

$$\nabla f_{\mathrm{DPO}} = -(1 - \sigma(q_\beta))\nabla q_\beta, \quad \nabla g_{\mathrm{SFT}} = -\nabla\pi/\pi. \quad (15)$$

Figure 5: Comparisons of $\delta(x)$ and $\nabla_y f(x, y_g^*(x))$ during PBGD-Free updates. The Lipschitz constant $\ell_{f,0} = \max_x \|\nabla_y f(x, y_g^*(x))\|$ is large but $\delta(x)$ is small.

While the Lipschitz constant for this problem is large, it satisfies the flatness condition with small $\delta$. To illustrate, we revisit the example in Figure 2; see the detailed setting in Appendix C.1. As in Figure 5, the flatness constant $\delta(x) \leq 0.0003$ in the blue line is small throughout optimization for $c = 0.5$ and $\alpha = 1.5$, despite a large Lipschitz constant in red. This confirms that the loss landscape analysis under the Lipschitz condition is not tight, as $l_{f,0}$ remains non-negligible even in local neighborhoods, whereas the flatness condition allows for a tighter analysis. The small $\delta(x)$ values along the PBGD-Free trajectory validate that *the PEFT problem* (3) *satisfies the flatness condition*, which inspired us to establish the enhanced analysis of the PBGD-Free algorithm under the flatness condition. In real-world PEFT problems, e.g. ones in Section 4, $\delta(x)$ in (12) is typically small because the distance between $y_g^*(x)$ and $y_\gamma^*(x)$ are non-negligible, whereas their impact on $f_{\mathrm{DPO}}$ is marginal.

### 3.3 Convergence analysis for the PBGD-Free algorithm

As shown in (9), the first term is the major bottleneck of the divergence issue of the PBGD-Free algorithm under the Lipschitz condition in Proposition 2. The key to establishing the convergence guarantee for the PBGD-Free algorithm is the tighter bound of $\|y_g^*(x) - y_\gamma^*(x)\|$ and $\|\phi(x) - F_\gamma(x)\|$ under the flatness condition, compared to the results in Lemma 1. We highlight the results as follows.

**Lemma 1** (**Tighter analysis on function value gap**). *Suppose Assumption 2.(iii) holds. For fixed $x$, suppose $f(x, \cdot)$ is $(\delta, \alpha)$-flat at any $y_g^*(x) \in S_g^*(x)$ with $\alpha \in (1, 1.5]$. Then, there exists $\gamma^* > 0$ such that for $\gamma \geq \gamma^*$, we have*

$$\|\phi(x) - F_\gamma(x)\| = \mathcal{O}(\gamma^{-\frac{\alpha}{2-\alpha}} + \delta), \quad \text{and} \quad d_{S_\gamma^*(x)}(y_g), d_{S_g^*(x)}(y_\gamma) = \mathcal{O}\left(\gamma^{-\frac{1}{2-\alpha}} + \delta^{\frac{1}{2}}\gamma^{-\frac{1}{2}}\right), \quad (16)$$

*for any $y_g \in S_g^*(x)$ and $y_\gamma \in S_\gamma^*(x)$.*

The proof of Lemma 1 is available at Appendix B.3. When $\delta$ is smaller than target accuracy $\epsilon$, achieving $\|\phi(x) - F_\gamma(x)\|, \|y_g^*(x) - y_\gamma^*(x)\|^2 = \mathcal{O}(\epsilon)$ only requires $\gamma = \mathcal{O}(\epsilon^{-\frac{2-\alpha}{2}})$, which is

strictly smaller than the choice of $\gamma = \mathcal{O}(\epsilon^{-\frac{1}{2}})$ in previous literature [78, 10, 45, 44]. This also aligns with common practice, where the penalty constant $\gamma$ does not need to be excessively large. For instance, the UL objective in Example 2 is $(10^{-3}, 1.1)$-flat and therefore choosing $\gamma = 15$ gives desired accuracy, supporting the rule of thumb: $\gamma \approx 15$ is a reasonable choice. In Figure 4, we also show that our bound under the flatness condition in Lemma 1 is tighter than the one under the Lipschitz condition in Proposition 1 for the representation learning PEFT (3).

Since Lemma 1 provides a per-iterate bound with fixed $x$, the next step is to analyze the Lipschitz continuity of the flatness constant $\delta(x)$ with respect to $x$, enabling a uniform bound across iterations.

**Lemma 2** (**Lipschitz continuity of flatness constant** $\delta(x)$)**.** *Suppose Assumption 2 holds. Then fixing some $c \geq 0$ and $\alpha \in (1, 2)$, there exists some trajectory of $y_g^*(x)$, $y_\gamma^*(x)$ such that the flatness constant of $f(x, \cdot)$, $\delta(x)$ defined in (12), is $\mathcal{O}(c\gamma^{-(\alpha-1)})$-Lipschitz-continuous in $x$.*

The proof of the Lemma 2 is available in Appendix B.4. However, Lemma 1 and Lemma 2 only enable the convergence of PBGD-Free to the stationary point of the penalized objective $F_\gamma(x)$. We next establish the approximate equivalence of the stationary points to the original BLO problem (1).

**Lemma 3** (**Approximate equivalence of stationary points**)**.** *Suppose Assumption 2 holds. Let $x^*$ be an $\epsilon$ stationary point of $F_\gamma(x^*)$ and suppose $f(x^*, \cdot)$ is $(\delta, \alpha)$-flat at any $y_g^*(x^*) \in S_g^*(x^*)$ with $\alpha \in (1, 1.5]$ and $\delta \leq \mathcal{O}(\epsilon^{\frac{\alpha}{2}})$. Then there exists $\gamma^* = \mathcal{O}(\epsilon^{-\frac{2-\alpha}{2}})$ and $y_\gamma \in S_\gamma^*(x^*)$ such that for $\gamma \geq \gamma^*$, $(x^*, y_\gamma)$ is the $\mathcal{O}(\epsilon)$ stationary point of the original BLO problem (1).*

The proof of the Lemma 3 is available in Appendix B.5, which generalizes the definition of stationary condition for (1) [91, 105] and its relations to that of the penalty problem [78, 45, 105] under flatness condition instead of Lipschitz continuity in Assumption 1. In this way, building upon Lemma 1 and Lemma 2, the convergence result for PBGD-Free in Algorithm 1 is stated as follows.

**Theorem 3** (**Convergence of PBGD-Free**)**.** *Suppose Assumption 2 holds, and for all $x_t$ on the trajectory, $f(x_t, \cdot)$ is $(\delta(x_t), \alpha)$-flat at all $y_g(x_t) \in S_g^*(x_t)$ with the same $\alpha \in (1, 1.5]$ and modulus $c = \mathcal{O}(1)$. For iterations generated by Algorithm 1 with $K = 1$ and step size $\eta \leq l_{F,1}^{-1}$, where $l_{F,1}$ is the smoothness constant of $F_\gamma(x)$, and suppose for target accuracy $\epsilon$, there exists $\delta$ such that $\frac{1}{T} \sum_{t=0}^{T-1} \delta(x_t) \leq \delta$, then by choosing $\gamma = \mathcal{O}(\delta^{-\frac{2-\alpha}{2}})$,*

$$\frac{1}{T} \sum_{t=0}^{T-1} \|\nabla F_\gamma(x_t)\|^2 \leq \mathcal{O}(T^{-1} + \delta^{\frac{2(\alpha-1)}{\alpha}}). \tag{17}$$

The proof of Theorem 3 is provided in Appendix B.6. Here, the smoothness constant $l_{F,1}$ is not scalable with $\gamma$ [11], therefore leading to a constant step size choice. Theorem 3 establishes the convergence rate of the fully-single-loop version of PBGD-Free in Algorithm 1. The result shows that the algorithm converges to the neighborhood of a stationary point for $F_\gamma(x)$, where the stationary gap is controlled by the flatness parameter $(\delta, \alpha)$. Specifically, for a $(\delta, \alpha)$-flat function with $\alpha \in (1, 1.5)$, the convergence error scales as $\mathcal{O}(\delta^{\frac{2(\alpha-1)}{\alpha}})$, ensuring that the suboptimality gap remains small. For instance, for the PEFT problem in (3), $\delta(x)$ is often negligible, as per the discussion in Section 3.2. Moreover, the method follows a single-loop update scheme, which is computationally more efficient than other fully first-order methods [44, 45, 105, 78, 10], as elaborated in the Appendix B.7. A comparison of the proposed algorithm with state-of-the-art fully first-order BLO methods is provided in Table 1.

## 4 Numerical Experiments

In this section, we empirically validate our theoretical results through experiments on real-world tasks. In the main paper, we will focus on the LLM PEFT problem (3). Additional experiments, including fair representation learning problem on the NLSY-7k dataset [73, 80], and BiDORA fine-tuning [68], are provided in Appendix C.

### 4.1 Representation learning based LLM PEFT and its flatness

SFT enhances pre-trained LLMs for downstream task adaptation, whereas DPO aligns them with human preferences. A straightforward way to achieve both goals is to sequentially optimize the

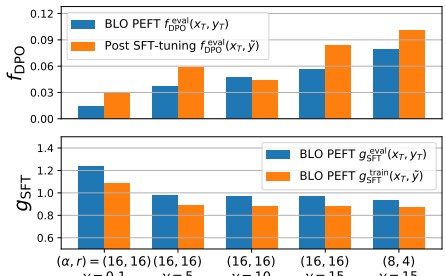

Figure 6: Ablation study on penalty $\gamma$ and LoRA configuration [35] for PYTHIA-1b [6].

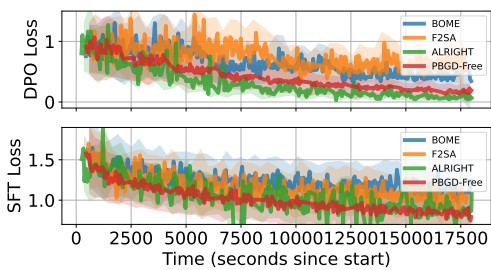

Figure 7: Train losses vs. time for different algorithms in solving (3) (or bi-objective learning for ALRIGHT) on LLAMA-3-3B [31].

| Methods | $f_{\text{DPO}}^{\text{eval}}(x_T, y_T)$ | $g_{\text{SFT}}^{\text{eval}}(x_T, y_T)$ | $f_{\text{DPO}}^{\text{eval}}(x_T, \tilde{y})$ | $g_{\text{SFT}}^{\text{train}}(x_T, \tilde{y})$ |
|---|---|---|---|---|
| V-PBGD [45] | 0.818 | 1.0309 | 0.8423 | 0.9533 |
| BOME [105] | 0.8332 | 1.1552 | 0.8402 | 0.9842 |
| ALRIGHT [24] | 0.8055 | 0.8656 | 0.8201 | 0.7855 |
| **PBGD-Free** | **0.7837** | **0.8516** | **0.8088** | **0.6688** |

Table 2: Comparison of different algorithms for PEFT LLAMA-3-3B [31]. Results show the DPO Loss ↓, SFT Loss ↓ for both the outcome $(x_T, y_T)$ trained on solving (3) for different methods or and the outcome $(x_T, \tilde{y})$ from post-SFT-tuning on another dataset with fixed-backbone, using the same dataset fixed time of training for each.

two objectives. However, this often leads to catastrophic forgetting [24], where applying DPO after SFT overwrites task-specific knowledge. To address this, we adopt the bilevel framework (3), which prioritizes SFT at the LL to create a more reliable base model and applies DPO at the UL to guide the human preference alignment. This hierarchical formulation effectively optimizes DPO conditioned on a near-optimal SFT, thereby preserving downstream task performance. Moreover, this design is natural, as user preferences are generally consistent across tasks due to underlying psychological regularities. Additionally, the proposed BLO framework aligns with the post-SFT paradigm, where DPO is fine-tuned from a pre-trained model and SFT is applied to downstream tasks. In practical settings, it is common to fine-tune only a lightweight head while keeping the backbone fixed or lightly updated [67, 106, 72].

In this paper, we adopt a decomposition of LLM into a backbone model $x$ (e.g., attention weights) and an output head $y$ to formulate a BLO PEFT framework (3). Our method conforms to the PEFT practice by allowing the head to specialize in SFT tasks while training the backbone through DPO to capture generalizable preference representations. In our experiments, we adopt the low rank adaptation (LoRA) [35] to the backbone $x$ for PEFT on LLAMA-3-3B [31] and PYTHIA-1b [6], using the `Dahoas/rm-hh-rlhf` dataset for DPO and the `OpenOrca` dataset [55] for SFT. Our code is adapted from the bilevel LLM post-training library `https://github.com/Post-LLM/BIPOST` and experiment details are referred to Appendix C.3. As preliminarily demonstrated in Figure 5, this *BLO PEFT problem* in (3) features flatness (small $\delta$), which is further corroborated by the observation in experiment that the LL solution $y_g^*(x)$ and $y_\gamma^*(x)$ have $\ell_2$- distance greater than 1, suggesting a negligible flatness constant $\delta(x)$ by (12).

### 4.2 Ablation study and main experimental results for the PEFT problem (3)

In this experiment, we consider evaluating methods on both **S1)** *BLO PEFT learning phase via* (3) to obtain a preference backbone $x$, and **S2)** *post-SFT tuning on a new dataset* with the obtained preference backbone model $x$, to verify the representation quality and transferability of the backbone.

We first conduct an ablation study on the PYTHIA-1b to test the impact of the penalty constant $\gamma$ and LoRA configuration on the PBGD-Free method. We report the DPO and SFT loss under different settings for both $(x_T, y_T)$ learned from **S1)** and $(x_T, \tilde{y})$ from **S2)** in Figure 6.

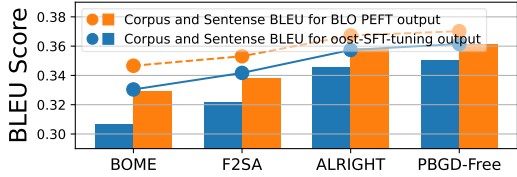

Figure 8: BLEU-4 Corpus and BLEU-4 Sentence Score (↑) for different algorithms for PEFT on LLAMA-3-3B [31].

**Trade-off between DPO and SFT under different $\gamma$.** According to Figure 6, increasing $\gamma$ degrades DPO performance while improving SFT for **S1)**, indicating that a larger $\gamma$ provides better LL optimality. Notably, the SFT improvement beyond $\gamma = 10$ is marginal for **S1)**, while the DPO performance significantly deteriorates, suggesting that $\gamma \approx 10$ offers the best balance as our theory predicts.

**Faster convergence over BLO baselines and stable training over bi-objective.** Since second-order BLO algorithms are inefficient in large-scale LLM training, we consider first-order methods F$^2$SA [45] and BOME [105] as BLO baselines. As shown in Figure 7, PBGD-Free converges faster than the BLO baselines. We additionally compare with ALRIGHT [24], an effective bi-objective algorithm, to validate the representation capability of *BLO PEFT* (3) formulation. ALRIGHT [24] exhibits less stability during training (Figure 7), likely due to alternating between DPO and SFT objectives.

**Transferability of preference backbone and strong SFT performance.** Compared with **S1)**, PBGD-Free in Figure 6 shows enhanced SFT with comparable DPO performance on **S2)**, suggesting it learns a transferable preference backbone $x$ through BLO (3). Table 2 further quantifies these findings for other baselines, demonstrating that PBGD-Free achieves superior DPO and SFT performance. Notably, the backbone model $x$ obtained by PBGD-Free attains the lowest SFT and DPO loss on **S2)**, verifying the transferability of PBGD-Free. To further evaluate the quality of generated output, Figure 8 corroborates the SFT performance using the evaluation metrics BLEU score [65], where our method outperforms all baselines, further justifying its superiority in learning a good representation. More experimental results, including semantic analysis (Table 6) are provided in Appendix C.3.

## 5 Concluding remarks

In this paper, we propose PBGD-Free, a penalty-based method for efficiently solving the nonconvex BLO problem without solving the value-function subproblem of $y_g^*(x)$. We first show that, under a general Lipschitz condition, the convergence of PBGD-Free has a constant lower bound by the Lipschitz constant, which does not vanish unless the Lipschitz constant is sufficiently small. Motivated by empirical findings in representation learning, we then introduce a Hölder-like condition and prove that, when its constant is sufficiently small, the fully single-loop PBGD-Free algorithm achieves an iteration complexity of $\mathcal{O}(\epsilon^{-1})$. We further demonstrate that this Hölder-like condition with a small constant is strictly weaker than the small Lipschitz condition, and we verify this condition in representation-learning-based LLM PEFT, fair representation learning, and BiDORA fine-tuning. Numerical experiments in the above problems demonstrate that the PBGD-Free algorithm is computationally efficient and can outperform the existing baselines across all three applications.

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

# Supplementary Material for "Beyond Value Functions: Single-Loop Bilevel Optimization under Flatness Conditions"

## Table of Contents

## A   Preliminaries

**Notations.** We define $v(x) = \min_z g(x, z)$ and $v_\gamma(x) = \min_y \gamma^{-1} f(x, y) + g(x, y)$. We denote $S_g^*(x) = \arg\min_z g(x, z)$, $S_\gamma^*(x) = \arg\min_y \gamma^{-1} f(x, y) + g(x, y)$, $d_S(y) := \min_{z \in S} \|y - z\|$.

**Definition 2** (Lipschitz Continuity and Smoothness). *We say a function $f(x, y)$ is $l_{f,0}$-Lipschitz if*

$$\|f(x_1, y_1) - f(x_2, y_2)\| \le l_{f,1} \|[x_1; y_1] - [x_2; y_2]\|, \quad \forall (x_1, y_1), (x_2, y_2) \tag{18}$$

*If $f$ is differentiable, we say $f$ is $l_{f,1}$-smooth on if $\nabla f$ is $l_{f,1}$-Lipschitz, i.e. $\quad \forall (x_1, y_1), (x_2, y_2)$:*

$$\|[\nabla_x f(x_1, y_1) - \nabla_x f(x_2, y_2); \nabla_y f(x_1, y_1) - \nabla_y f(x_2, y_2)]\| \le l_{f,1} \|[x_1; y_1] - [x_2; y_2]\|. \tag{19}$$

**Definition 3** (PL condition). *We say $g(x, y)$ satisfies $\mu$-Polyak-Łojasiewicz (PL) condition in $y$ if*

$$\|\nabla_y g(x, y)\| \ge 2\mu(g(x, y) - v(x)). \tag{20}$$

**Lemma 4** ([41, Theorem 2]). *If $g(x, y)$ is $\ell_{g,1}$-Lipschitz smooth and PL in $y$ with $\mu_g$, then it satisfies the error bound (EB) condition with $\mu_g$, i.e.*

$$\|\nabla_y g(x, y)\| \ge \mu_g d_{S_g^*(x)}(y). \tag{21}$$

*Moreover, it also satisfies the quadratic growth (QG) condition with $\mu_g$, i.e.*

$$g(x, y) - v(x) \ge \frac{\mu_g}{2} d_{S_g^*(x)}(y)^2. \tag{22}$$

*Conversely, if $g(x, y)$ is $\ell_{g,1}$-Lipschitz smooth and satisfies EB with $\mu_g$, then it is PL in $y$ with $\mu_g / \ell_{g,1}$.*

**Proposition 4** (Complete version to Proposition 1 [78, 45]). *Under Assumption 1–2, for any $x$, there is*

$$\|F_\gamma(x) - \phi(x)\| \le \mathcal{O}(\|\nabla_y f(x, y_g^*(x))\|^2 \mu^{-1} \gamma^{-1}) = \mathcal{O}(l_{f,0}^2 \mu^{-1} \gamma^{-1}). \tag{23}$$

*Additionally, for any $y_g^*(x) \in S_g^*(x)$, $y_\gamma^*(x) \in S_\gamma^*(x)$,*

$$d_{S_\gamma^*(x)}(y_g^*(x)), d_{S_g^*(x)}(y_\gamma^*(x)) \le \Omega(\|\nabla_y f(x, y_g^*(x))\| \mu^{-1} \gamma^{-1}) = \Omega(l_{f,0} \mu^{-1} \gamma^{-1}). \tag{24}$$

*Moreover, for $y_g^*(x) = \arg\min_{z \in S_g^*(x)} f(x, z)$, there is*

$$\|\nabla \phi(x) - \nabla F_\gamma(x)\| = \mathcal{O}\left(d_{S_\gamma^*(x)}(y_g^*(x))\right) = \mathcal{O}(l_{f,0} \mu^{-1} \gamma^{-1}).$$

## A.1 Proof of Proposition 2

First of all, Algorithm 1 can be viewed as a biased PBGD algorithm with bias being

$$
\begin{aligned}
\|b_t\| =&\|\nabla F_\gamma(x_t) - \nabla_x f(x_t, y_t^\gamma)\|\\
\stackrel{(a)}{=}&\|\nabla_x f(x_t, y_\gamma^*(x)) + \gamma(\nabla_x g(x_t, y_\gamma^*(x_t)) - \nabla_x g(x_t, y_g^*(x_t))) - \nabla_x f(x_t, y_t^\gamma)\|\\
\stackrel{(b)}{\leq}&\|\nabla_x f(x_t, y_\gamma^*(x)) - \nabla_x f(x_t, y_t^\gamma)\| + \gamma\|\nabla_x g(x_t, y_\gamma^*(x_t)) - \nabla_x g(x_t, y_g^*(x_t))\|\\
\stackrel{(c)}{\leq}&l_{f,1}\|y_\gamma^*(x_t) - y_t^\gamma\| + \gamma l_{g,1}\|y_\gamma^*(x_t) - y_g^*(x_t)\|\\
\stackrel{(d)}{\leq}&l_{f,1}\sqrt{\gamma^{-1} f(x_t, y_t^\gamma) - v_\gamma(x_t)} + \mathcal{O}(\|\nabla_y f(x, y_g^*(x_t))\|)\\
\stackrel{(e)}{\leq}&l_{f,1}\sqrt{(1 - \eta^\gamma\mu)^K(\gamma^{-1} f(x_t, y_{t-1}^\gamma) - v_\gamma(x_t))} + \mathcal{O}(\|\nabla_y f(x, y_g^*(x_t))\|)
\end{aligned}
$$

where (a) is by plugging in $\nabla F_\gamma(x_t)$ in (5), this holds for arbitrary $y_g^*(x)$, $y_\gamma(x)$ as solutions to problems in (5); (b) follows triangle-inequality; (c) uses the smoothness of $f$ and $g$; the first term in (d) is obtained by the QG property as ensured by PL condition and smoothness as per Lemma 4, via choosing $y_\gamma^*(x_t) = \arg\min_{y \in S_\gamma^*(x_t)}\|y - y_t^\gamma\|$, and the second term follows Proposition 4 by choosing $y_g^*(x) = \arg\min_{z \in S_g^*(x_t)}\|y_\gamma^*(x_t) - z\|$; and (e) follows the linear convergence result of PL function [41] as $y_t^\gamma$ is the results from $K$-step inner update starting at $y_{t-1}^\gamma$. In this way, when taking $K$ sufficiently large, there is $\|b_t\| \leq \mathcal{O}(\|\nabla_y f(x, y_g^*(x_t))\|) = \mathcal{O}(l_{f,0})$.

Moreover, according to [11], $F_\gamma(x)$ is $\mathcal{O}(1)$-smooth. Therefore,

$$
\frac{1}{T}\sum_{t=0}^{T-1}\|\nabla F_\gamma(x_t)\|^2 \leq \mathcal{O}(T^{-1}) + \sum_{t=0}^{T-1}\|b_t\|^2
$$

$$
\leq \mathcal{O}(T^{-1}) + \mathcal{O}\Big(\frac{1}{T}\sum_{t=0}^{T}\|\nabla_y f(x, y_g^*(x_t))\|^2\Big) = \mathcal{O}(T^{-1}) + \mathcal{O}(l_{f,0}^2). \quad (25)
$$

In this way, for sufficiently large $T$, $\frac{1}{T}\sum_{t=0}^{T-1}\|\nabla F_\gamma(x)\|^2 \leq \mathcal{O}(l_{f,0}^2)$.

Next, we will prove the lower bound of Algorithm 1 by constructing a counterexample, and show that the upper bound is tight. Consider $f(x, y) = x^2 + l_{f,0}y$ and $g(x, y) = (y - x + 1)^2$. In this problem, $\nabla F_\gamma(x) = 2x + l_{f,0}$ while $\nabla_x f(x, y_\gamma^*(x)) = 2x$. Using fixed stepsize $\eta$, $x_{t+1} = x_t - \eta x_t = (1 - \eta)x_t$, implying $\|\nabla_x f(x_{t+1}, y_\gamma^*(x_{t+1}))\| = \|2x_{t+1}\| = 2(1 - \eta)\|x_t\| = 2(1 - \eta)^{t+1}\|x_0\|$. Therefore, for arbitrary small $\epsilon > 0$, there exists some $T_0 = \mathcal{O}(\ln(\epsilon^{-1}))$ such that Algorithm 1 converges to $\|x_t\| < \epsilon$ for all $t \geq T_0$, whereas $\nabla F_\gamma(x_t) = l_{f,0}$. In this way, we have

$$
\frac{1}{T - T_0}\sum_{t=T_0}^{T}\|\nabla F_\gamma(x_t)\|^2 = \mathcal{O}(\epsilon) + l_{f,0}^2. \quad (26)
$$

This indicates $\Omega(l_{f,0}^2)$ is a tight bound.

# B   Improved Analysis under Flatness

## B.1   Proof of Observation 1

For $\|y - y_g^*(x)\| \geq 1$, by the $l_{f,0} = \delta^{\frac{1}{\alpha}}$-Lipschitzness of $f(x, y)$ in $y$, there is

$$
\|f(x, y) - f(x, y_g^*(x))\| \leq l_{f,0}\|y - y_g^*(x)\| \leq \delta^{\frac{1}{\alpha}}\|y - y_g^*(x)\|^\alpha. \quad (27)
$$

For small $\|y - y_g^*(x)\| < 1$, Taylor's expansion gives

$$
f(x, y) - f(x, y_g^*(x)) = \langle\nabla_y f(x, y_g^*(x)), y - y_g^*(x)\rangle + R(x, y). \quad (28)
$$

Here, $R(x, y)$ is a remainder. By Hölder-Continuous Gradient Condition [5], [62, Section 2], which is implied by the smoothness, there exists some $0 \leq c \leq l_{f,1}/2$, $1 < \alpha < 2$ such that $\|R(x, y)\| \leq c\|y - y_g^*(x)\|^\alpha$. By Cauchy-Schwartz's inequality, there is

$$
\begin{aligned}
\|\langle \nabla_y f(x, y_g^*(x)), y - y_g^*(x)\rangle\| &\leq \|\nabla_y f(x, y_g^*(x))\|\|y - y_g^*(x)\| \\
&\leq \delta^{\frac{1}{\alpha}}\|y - y_g^*(x)\| \\
&\leq \delta + \|y - y_g^*(x)\|^\alpha
\end{aligned}
\tag{29}
$$

where the last inequality holds as for $a, b \in (0, 1)$ and $\alpha \in (1, 2)$, there is $ab \leq \max\{a, b\}^2 \leq \max\{a, b\}^\alpha \leq a^\alpha + b^\alpha$ and here $a = \delta^{1/\alpha}$ and $b = \|y - y_g^*(x)\|$. The observation therefore, holds.

## B.2  Detailed example for Observation 2

The following example visualizes Observation 2.

**Example 2.** *We consider the LL objective $g(x, y) = (y - x)^2$ and the UL objective*

$$
f(x, y) = (\sin(y - x) + 2)|y - x|^2 + 10\exp\left(-\frac{(y - x)^2}{2(0.005)^2}\right)\sin(100(y - x)).
$$

*The LL problem $g(x, y)$ is strongly convex in $y$, with $y_g^*(x) = x$. Therefore, $\nabla_y f(x, y_g^*(x)) = 1000$ is extremely large, which leads to a loose upper bound for $\|y_g^*(x) - y_\gamma^*(x)\|$ or $\|\phi(x) - F_\gamma(x)\|$ following Lemma 1. However, this problem is $(3e^{-3}, 1.1)$-flat with $c = 5$ at $y_g^*(x) = x$ for $x \in [-10, 10]$. As shown in Figure 3, $f(x, \cdot)$ exhibits fluctuations around $y_g^*(x)$ while remaining relatively stable elsewhere. This shows that flatness is weaker than requiring small $\|\nabla_y f(x, y)\|$.*

## B.3  Proof of Lemma 1

*Proof.* For any $y_g^*(x) \in S_g^*(x)$, $y_\gamma^*(x) \in S_\gamma^*(x)$, there is

$$
\begin{aligned}
&\gamma^{-1}f(x, y_\gamma^*(x)) + g(x, y_\gamma^*(x)) \leq \gamma^{-1}f(x, y_g^*(x)) + g(x, y_g^*(x)) \\
\Rightarrow\quad &\gamma^{-1}f(x, y_\gamma^*(x)) + g(x, y_\gamma^*(x)) - v(x) \leq \gamma^{-1}f(x, y_g^*(x)) + g(x, y_g^*(x)) - v(x) \\
\Rightarrow\quad &\gamma^{-1}f(x, y_\gamma^*(x)) + g(x, y_\gamma^*(x)) - v(x) \leq \gamma^{-1}f(x, y_g^*(x)) \\
&\qquad\qquad\Rightarrow\quad f(x, y_\gamma^*(x)) \leq f(x, y_g^*(x)).
\end{aligned}
\tag{30}
$$

In this way, according to the definition of $\phi(x)$ and $F_\gamma(x)$, we have

$$
\begin{aligned}
\|\phi(x) - F_\gamma(x)\| &= \min_{z \in S_g^*(x)} f(x, z) - \big(f(x, y_\gamma^*(x)) + \gamma(g(x, y_\gamma^*(x)) - v(x))\big) \\
&\overset{(a)}{\leq} f(x, y_g^*(x)) - \big(f(x, y_\gamma^*(x)) + \gamma(g(x, y_\gamma^*(x)) - v(x))\big) \\
&\overset{(b)}{\leq} f(x, y_g^*(x)) - \left(f(x, y_\gamma^*(x)) + \gamma\frac{\mu_g}{2}\|y_g^*(x) - y_\gamma^*(x)\|^2\right) \\
&\overset{(c)}{=} \|f(x, y_g^*(x)) - f(x, y_\gamma^*(x))\| - \gamma\frac{\mu_g}{2}\|y_g^*(x) - y_\gamma^*(x)\|^2 \\
&\overset{(d)}{\leq} c\|y_g^*(x) - y_\gamma^*(x)\|^\alpha + \delta - \gamma\frac{\mu}{2}\|y_g^*(x) - y_\gamma^*(x)\|^2 \\
&\overset{(e)}{\leq} \max_{z:z\geq 0} cz^\alpha - \gamma\frac{\mu}{2}z^2 + \delta \\
&\overset{(f)}{=} c^{\frac{2}{2-\alpha}}(2\alpha)^{\frac{\alpha}{2-\alpha}}(1 - \alpha/2)(\mu\gamma)^{-\frac{\alpha}{2-\alpha}} + \delta = \mathcal{O}(\gamma^{-\frac{\alpha}{2-\alpha}} + \delta)
\end{aligned}
\tag{31}
$$

Here, (a) holds for arbitrary $y_g^*(x) \in S_g^*(x), y_\gamma^*(x) \in S_\gamma^*(x)$ by (30); (b) is from the $\mu_g$ quadratic growth condition of $g(x, \cdot)$ which is implied by $\mu_g$-PL according to Lemma 4, via choosing $y_g^*(x) = \arg\min_{z \in S_g^*(x)}\|z - y_\gamma^*(x)\|$; (c) again uses (30); (d) follows the flatness of $f(x, y)$ at $y_g^*(x)$, (e) is by formulating the problem as a maximization problem over $z = \|y_g^*(x) - y_\gamma^*(x)\|$; and (f) is the solution to this polynomial problem. Therefore, the first part is proved.

For the second part, as $\frac{1}{\gamma}f(x,\cdot) + g(x,\cdot)$ being $\mu$-PL for $\gamma > \gamma^*$, it is also $\mu$-QG by Lemma 4. In this way, fixed any $\gamma > \gamma^*$, for any $y_\gamma^*(x) \in S_\gamma^*(x)$ and any $y_g^*(x) \in S_g^*(x)$, there is

$$\gamma\left(\left(\frac{1}{\gamma}f(x,y_g^*(x)) + g(x,y_g^*(x))\right) - \left(\frac{1}{\gamma}f(x,y_\gamma^*(x)) + g(x,y_\gamma^*(x))\right)\right) \geq \gamma\frac{\mu}{2}d_{S_\gamma^*(x)}^2(y_g^*(x)). \tag{32}$$

Moreover, following steps (a)-(d) as in (31), there is

$$\text{left of (32)} = \left(f(x,y_g^*(x)) + \gamma g(x,y_g^*(x)) - \gamma v(x)\right) - \left(f(x,y_\gamma^*(x)) + \gamma g(x,y_\gamma^*(x)) - \gamma v(x)\right)$$
$$= f(x,y_g^*(x)) - f(x,y_\gamma^*(x)) - \gamma\left(g(x,y_\gamma^*(x)) - v(x)\right)$$
$$\leq c\|y_g^*(x) - y_\gamma^*(x)\|^\alpha + \delta - \gamma\frac{\mu}{2}d_{S_\gamma^*(x)}^2(y_\gamma^*(x)) \tag{33}$$

Combining (32) and the above, there is

$$c\|y_g^*(x) - y_\gamma^*(x)\|^\alpha + \delta - \gamma\frac{\mu}{2}d_{S_g^*(x)}^2(y_\gamma^*(x)) \geq \gamma\frac{\mu}{2}d_{S_\gamma^*(x)}^2(y_g^*(x)) \tag{34}$$

for any $y_g^*(x) \in S_g^*(x)$ and $y_\gamma^*(x) \in S_\gamma^*(x)$. In this way, for any $y_g^*(x) \in S_g^*(x)$, choose $y_\gamma^*(x) = \arg\min_{y \in S_\gamma^*(x)}\|y - y_g^*(x)\|$, there is

$$cd_{S_\gamma^*(x)}^\alpha(y_g^*(x)) + \delta \geq \gamma\frac{\mu}{2}d_{S_\gamma^*(x)}^2(y_g^*(x)) \tag{35}$$

Similarly, for any $y_\gamma^*(x) \in S_\gamma^*(x)$, choose $y_g^*(x) = \arg\min_{z \in S_g^*(x)}\|z - y_\gamma^*(x)\|$, there is

$$cd_{S_g^*(x)}^\alpha(y_\gamma^*(x)) + \delta \geq \gamma\frac{\mu}{2}d_{S_g^*(x)}^2(y_\gamma^*(x)). \tag{36}$$

For simplicity, denote $x = d_{S_g^*(x)}(y_\gamma^*(x))$ (or $x = d_{S_\gamma^*(x)}(y_g^*(x))$), there is

$$x^{2-\alpha} \leq 2c\mu^{-1}\gamma^{-1} + 2\delta\mu^{-1}\gamma^{-1}x^{-\alpha}. \tag{37}$$

As $\alpha \in (1,1.5]$, for $x \geq \sqrt{\frac{\delta}{\gamma}}$,

$$x^{2-\alpha} \leq 2c\mu^{-1}\gamma^{-1} + 2\delta\mu^{-1}\gamma^{-1}\left(\frac{\delta}{\gamma}\right)^{-\frac{\alpha}{2}}. \tag{38}$$

Since $|a+b|^p \leq 2^{p-1}(|a|^p + |b|^p)$ for all $p \geq 1$ (as $|\cdot|^p$ is convex), there is

$$x = (x^{2-\alpha})^{\frac{1}{2-\alpha}} \leq \left(2c\mu^{-1}\gamma^{-1} + 2\delta\mu^{-1}\gamma^{-1}\left(\frac{\delta}{\gamma}\right)^{-\frac{\alpha}{2}}\right)^{\frac{1}{2-\alpha}}$$
$$\leq 2^{\frac{1}{2-\alpha}-1}\left((2c\mu^{-1})^{\frac{1}{2-\alpha}}\gamma^{-\frac{1}{2-\alpha}} + (2\mu^{-1})^{\frac{1}{2-\alpha}}\delta^{\frac{1}{2}}\gamma^{-\frac{1}{2}}\right) = Oc(\gamma^{-\frac{1}{2-\alpha}} + \delta^{\frac{1}{2}}\gamma^{-\frac{1}{2}}) \tag{39}$$

In this way, we can conclude the following to include the scenario that $x \leq \sqrt{\frac{\delta}{\gamma}}$.

$$x = \mathcal{O}(\gamma^{-\frac{1}{2-\alpha}} + \delta^{\frac{1}{2}}\gamma^{-\frac{1}{2}}). \tag{40}$$

$\square$

## B.4 Proof of Lemma 2

Define

$$\delta'(x) := |f(x,y_g^*(x)) - f(x,y_\gamma^*(x))| - c\|y_g^*(x) - y_\gamma^*(x)\|^\alpha$$
$$= f(x,y_g^*(x)) - f(x,y_\gamma^*(x)) - c\|y_g^*(x) - y_\gamma^*(x)\|^\alpha. \tag{41}$$

where the equality is from $f(x,y_g^*(x)) \geq f(x,y_\gamma^*(x))$ as per (30). We firstly show that $f(x,y_g^*(x)) - f(x,y_\gamma^*(x))$ and $c\|y_g^*(x) - y_\gamma^*(x)\|^\alpha$ are both Lipschitz continuous.

For $f(x, y_g^*(x)) - f(x, y_\gamma^*(x))$, according to [11, Lemma F.3], there is

$$\left\| \frac{\partial}{\partial x} \left[ f(x, y_g^*(x)) - f(x, y_\gamma^*(x)) \right] \right\|$$

$$= \left\| \nabla_x f(x, y_g^*(x)) - \nabla_x f(x, y_\gamma^*(x)) + \nabla_y f(x, y_g^*(x)) \nabla_{yy} g(x, y_g^*(x))^\dagger \nabla_{yx} g(x, y_g^*(x)) \right.$$
$$\left. - \nabla_y f(x, y_\gamma^*(x)) \left[ \gamma^{-1} \nabla_{yy} f(x, y_\gamma^*(x)) + \nabla_{yy} g(x, y_\gamma^*(x)) \right]^\dagger \right.$$
$$\left. \times \left[ \gamma^{-1} \nabla_{yx} f(x, y_\gamma^*(x)) + \nabla_{yx} g(x, y_\gamma^*(x)) \right] \right\|$$

$$\leq \left\| E_1 \right\| + \left\| \nabla_y f(x, y_g^*(x)) \nabla_{yy} g(x, y_g^*(x))^\dagger \nabla_{yx} g(x, y_g^*(x)) \right.$$
$$\left. - [\nabla_y f(x, y_g^*(x)) + E_2][\nabla_{yy} g(x, y_g^*(x)) + E_3]^\dagger [\nabla_{yx} g(x, y_g^*(x)) + E_4] \right\| \tag{42}$$

where the inequality is by the triangle inequality and by denoting

$$\begin{cases} E_1 = & \nabla_x f(x, y_\gamma^*(x)) - \nabla_x f(x, y_g^*(x)) \\ E_2 = & \nabla_y f(x, y_\gamma^*(x)) - \nabla_y f(x, y_g^*(x)) \\ E_3 = & \gamma^{-1} \nabla_{yy} f(x, y_\gamma^*(x)) + \nabla_{yy} g(x, y_\gamma^*(x)) - \nabla_{yy} g(x, y_g^*(x)) \\ E_4 = & \gamma^{-1} \nabla_{yx} f(x, y_\gamma^*(x)) + \nabla_{yx} g(x, y_\gamma^*(x)) - \nabla_{yx} g(x, y_g^*(x)) \end{cases} \tag{43}$$

By the smoothness of $f$, the Lipschitzness of $\nabla^2 g$ and by Proposition 1, we know that

$$\|E_1\|, \|E_2\|, \|E_3\|, \|E_4\| = \mathcal{O}(\gamma^{-1}). \tag{44}$$

Additionally, according to [89], we know

$$\|[\nabla_{yy} g(x, y_g^*(x)) + E_3]^\dagger - \nabla_{yy} g(x, y_g^*(x))^\dagger\| \leq \frac{1 + \sqrt{5}}{\mu} \|E_3\| = \mathcal{O}(\gamma^{-1}). \tag{45}$$

In this way, by the smoothness of $f$ and $g$, we know $\|\gamma^{-1} \nabla^2 f + \nabla^2 g\| \leq \gamma^{-1} l_{f,1} + l_{g,1}$ and therefore,

$$\left\| \frac{\partial}{\partial x} \left[ f(x, y_g^*(x)) - f(x, y_\gamma^*(x)) \right] \right\| \leq (42) \leq \mathcal{O}(\gamma^{-1}). \tag{46}$$

This shows that $f(x, y_g^*(x)) - f(x, y_\gamma^*(x))$ is Lipschitz-continuous.

Fix any $x$, denote arbitrary $y_g^*(x) \in S_g^*(x)$, $y_\gamma^*(x) \in S_\gamma^*(x)$, then for any $x' \in \mathcal{X}$, there exists some $y_g^*(x') \in S_g^*(x')$, $y_\gamma^*(x') \in S_\gamma^*(x')$ such that

$$\left| c\|y_g^*(x) - y_\gamma^*(x)\|^\alpha - c\|y_g^*(x') - y_\gamma^*(x')\|^\alpha \right|$$

$$\overset{(a)}{\leq} c \max_{z \in [\|y_g^*(x) - y_\gamma^*(x)\|, \|y_g^*(x') - y_\gamma^*(x')\|]} z^{\alpha-1} \left| \|y_g^*(x) - y_\gamma^*(x)\| - \|y_g^*(x') - y_\gamma^*(x')\| \right|$$

$$\overset{(b)}{\leq} \mathcal{O}(c\gamma^{-(\alpha-1)}) \left\| (y_g^*(x) - y_\gamma^*(x)) - (y_g^*(x') - y_\gamma^*(x')) \right\|$$

$$\overset{(c)}{\leq} \mathcal{O}(c\gamma^{-(\alpha-1)}) \left( \|y_g^*(x) - y_g^*(x')\| + \|y_\gamma^*(x) - y_\gamma^*(x')\| \right)$$

$$\overset{(d)}{=} \mathcal{O}(c\gamma^{-(\alpha-1)}) \|x - x'\| \tag{47}$$

where (a) follows the mean value theorem, as $|\cdot|^\alpha$ is continuous; (b) is from $\|y_g^*(x) - y_\gamma^*(x)\| = \mathcal{O}(\gamma^{-1})$, and the 1-Lipschitzness of the norm function; (c) uses triangle-inequality; and (d) is achieved by knowing that $y_g^*(x)$ and $y_\gamma^*(x)$ are, respectively, $L_g$, $L_\gamma$-Lipschitz for some constant $L_g$, $L_\gamma$ [78].

In this way, we can conclude that $\delta'(x)$ is $\mathcal{O}(c\gamma^{-(\alpha-1)})$ Lipschitz continuous. As $\delta(x)$ is a ReLu function works on $\delta'(x)$, it is also $\mathcal{O}(c\gamma^{-(\alpha-1)})$ Lipschitz continuous.

### B.5 Stationary relations under flatness condition

Similar to [91], we first derive the stationary conditions for the original BLO problem (1), under the flatness condition in Definition 1 instead of the Lipschitz continuity. Then we prove the stationary equivalence of the penalty problem with the original BLO problem (1).

### B.5.1 Stationary conditions for original BLO problem (1)

First, the original BLO problem can be equivalently written as its gradient based constrained form under LL PL condition as follows [105, 91].

$$\min_{x,y} \ f(x,y), \quad \text{s.t.} \quad \nabla_y g(x,y) = 0. \tag{48}$$

We aim to show that the Karush–Kuhn–Tucker conditions (KKT) conditions of (48) are necessary for the global optimality of the original BLO problem (1), thereby serving as its stationary conditions. Prior works [104, 102] have discussed that under the calmness condition, the KKT condition is a necessary optimality condition. Similar to [104, 102, 91, 13], we will prove that the calmness condition holds for (48) for our problem, even under our relaxed assumptions, so that KKT conditions, which our proposed algorithm converges to, are necessary for the optimality in the original BLO problem (1). Notably, the key difference is that the prior works require a global [91] or local [13] Lipschitz condition of the upper-level objective, while we prove this under a more relaxed flatness condition of the upper-level objective. This makes the result applicable to a much wider set of problems. We first review the definition of the calmness condition below.

**Definition 4 (Calmness, [18, Definition 6.4.1]).** *Let $(x^*, y^*)$ be the global minimizer of the constrained problem*

$$\min_{x,y} \ f(x,y) \quad \text{s.t.} \ f_c(x,y) = 0. \tag{49}$$

*where $f_c : \mathbb{R}^{d_x + d_y} \to \mathbb{R}^d$ and $d \geq 1$. If there exist positive $\epsilon$ and $M$ such that for any $q \in \mathbb{R}^d$ with $\|q\| \leq \epsilon$ and any $\|(x', y') - (x^*, y^*)\| \leq \epsilon$ which satisfies $f_c(x', y') + q = 0$, one has*

$$f(x', y') - f(x^*, y^*) + M\|q\| \geq 0 \tag{50}$$

*then the problem (49) is said to be calm with $M$ and $\epsilon$.*

We will prove a general version for establishing that the KKT conditions of problem (48) serve as the stationary conditions of the BLO problem (1), which only requires the UL objective to be continuously differentiable.

**Lemma 5.** *Suppose that $g(x, \cdot)$ satisfies the PL condition and is smooth, and $f(x, \cdot)$ is continuously differentiable. For the global minimizer $(x^*, y^*)$ of BLO problem in (1) (a.k.a (48)), then (48) is calm at its global minimizer $(x^*, y^*)$, and therefore, the KKT conditions of (48) are the necessary conditions of the global optimality in (1).*

*Proof.* First, for any $x$, since $f(x, \cdot)$ is continuously differentiable, then $f(x, \cdot)$ is locally Lipschitz continuous around any $y'$, i.e. there exists a neighborhood of $\mathbb{B}_\epsilon(y') := \{y : \|y - y'\| \leq \epsilon\}$ and constant $L := \max_{y \in \mathbb{B}_\epsilon(y')} \|\nabla_y f(x, y)\| < \infty$ such that $f(x, \cdot)$ is Lipschitz continuous with constant $L$ over $y \in \mathbb{B}_\epsilon(y')$.

Then consider $\forall q$ with $\|q\| \leq \epsilon$, and $\forall x' \in \mathbb{B}_\epsilon(x^*)$ and $y'$, s.t. $\nabla_y g(x', y') + q = 0$, then letting $y_q \in \text{Proj}_{S_g^*(x')}(y')$ and according to Lemma 4, one has

$$\epsilon \geq \|q\| = \|\nabla_y g(x', y')\| \geq \mu_g \|y' - y_q\|$$

Since $(x^*, y^*)$ solves (48) and $(x', y_q)$ is also feasible to (48), one has $f(x^*, y^*) \leq f(x', y_q)$. Thus,

$$f(x', y') - f(x^*, y^*) \geq f(x', y') - f(x', y_q) \overset{(a)}{\geq} -L\|y' - y_q\| \geq -L\|q\| \tag{51}$$

where $(a)$ is due to the local Lipschitz continuity of $f(x, \cdot)$ with $L := \max_{y \in \mathbb{B}_\epsilon(y')} \|\nabla_y f(x, y)\|$.

(51) justifies the calmness definition in (50) with $M := \frac{L}{\mu_g}$ and $\epsilon$. $\qquad\square$

Therefore, under Assumption 2, the smoothness of $f(x, \cdot)$ implies that $f(x, \cdot)$ is continuously differentiable so that Lemma 5 holds. Note that Lemma 5 also generalizes the results in [91, 13] by relaxing the global/local Lipschitz continuity assumption on $f(x, \cdot)$ via continuously differentiable (ensuring local Lipschitz continuity).

We then aim to prove in Lemma 3 that the stationary point of the penalty reformulation (2) is approximately the stationary point of the original BLO problem in (1) (i.e., the KKT point of (48)).

### B.5.2  Proof of Lemma 3

*Proof.* Let $x^*$ be the stationary point of the penalty problem (2), then we have $\|\nabla F_\gamma(x^*)\| \le \epsilon$. Then according to (5), for $\forall y_\gamma \in S^*_\gamma(x^*)$ and $\forall y_g \in S^*_g(x^*)$, we have

$$\|\nabla_x f(x^*, y_\gamma) + \gamma(\nabla_x g(x^*, y_\gamma) - \nabla_x g(x^*, y_g))\|^2 \le \epsilon \tag{52a}$$

$$\|\nabla_y f(x^*, y_\gamma) + \gamma(\nabla_y g(x^*, y_\gamma) - \nabla_y g(x^*, y_g))\|^2 \le \epsilon. \tag{52b}$$

We aim to prove that $(x^*, y_\gamma)$ is approximately the KKT point of (48), i.e. $\exists$ finite $w \in \mathbb{R}^{d_y}$, s.t.

$$\|\nabla_x f(x^*, y_\gamma) + \nabla_{xy} g(x^*, y_\gamma) w\|^2 \le \mathcal{O}(\epsilon) \tag{53a}$$

$$\|\nabla_y f(x^*, y_\gamma) + \nabla_{yy} g(x^*, y_\gamma) w\|^2 \le \mathcal{O}(\epsilon) \tag{53b}$$

$$\|\nabla_y g(x^*, y_\gamma)\|^2 \le \mathcal{O}(\epsilon). \tag{53c}$$

The approximate LL optimality in (53c) is earned by Lemma 1, which gives

$$d_{S^*_g(x)}(y_\gamma) = \mathcal{O}(\gamma^{-\frac{1}{2-\alpha}} + \delta^{\frac{1}{2}} \gamma^{-\frac{1}{2}}) = \mathcal{O}(\epsilon^{0.5} + \delta^{0.5} \epsilon^{\frac{2-\alpha}{4}})$$

when $\gamma = \mathcal{O}(\epsilon^{-\frac{2-\alpha}{2}})$. Therefore, when $\delta \le \epsilon^{\frac{\alpha}{2}}$, it holds that

$$\|\nabla_y g(x^*, y_\gamma)\|^2 \le \mathcal{O}(d^2_{S^*_g(x)}(y_\gamma)) = \mathcal{O}(\epsilon + \delta \epsilon^{\frac{2-\alpha}{2}}) \le \mathcal{O}(\epsilon) \tag{54}$$

where the first inequality is earned by the smoothness condition. Moreover, by Taylor expansion of (52) and letting $y_g = \operatorname{argmin}_{y \in S^*_g(x)} \|y_\gamma - y_g\|$, it holds that

$$\|\nabla_x f(x^*, y_\gamma) + \gamma \nabla_{xy} g(x^*, y_\gamma)(y_\gamma - y_g)\|^2 \le \epsilon + \mathcal{O}(\|y_\gamma - y_g\|^2) \le \mathcal{O}(\epsilon),$$

$$\|\nabla_y f(x^*, y_\gamma) + \gamma \nabla_{yy} g(x^*, y_\gamma)(y_\gamma - y_g)\|^2 \le \epsilon + \mathcal{O}(\|y_\gamma - y_g\|^2) \le \mathcal{O}(\epsilon).$$

where the last two inequalities are due to (54). Together with (54) and defining $w = \gamma(y_\gamma - y_g)$ with finite norm $\|w\| = \mathcal{O}(\epsilon^{-\frac{2-\alpha}{2}} \epsilon) = \mathcal{O}(\epsilon^{\frac{\alpha}{2}}) \le 1$, the point $(x^*, y_\gamma, w)$ satisfies the approximate KKT conditions in (53). $\square$

### B.6  Proof of Theorem 3

In the following, we start with a more general setting where $x$ is bounded in a domain $\mathcal{X}$ and the update of $x$ is conducted via projected gradient descent.

Denote the gradient approximate $g_t = \nabla_x f(x, y^\gamma_{t+1})$. According to smoothness, we have

$$
\begin{aligned}
F_\gamma(x_{t+1}) - F_\gamma(x_t) &\le \langle \nabla F_\gamma(x_t) - g_t + g_t, x_{t+1} - x_t \rangle + \frac{l_{F,1}}{2} \|x_{t+1} - x_t\|^2 \\
&\le -\frac{1}{\eta} \|x_{t+1} - x_t\|^2 + \frac{1}{2\eta} \|x_{t+1} - x_t\|^2 + \|\nabla F_\gamma(x_{t+1}) - g_t\| \|x_{t+1} - x_t\| \\
&\le -\frac{1}{2\eta} \|x_{t+1} - x_t\|^2 + \frac{1}{4\eta} \|x_{t+1} - x_t\|^2 + \eta \|\nabla F_\gamma(x_{t+1}) - g_t\|^2 \\
&= -\frac{1}{4\eta} \|x_{t+1} - x_t\|^2 + \eta \|\nabla F_\gamma(x_{t+1}) - g_t\|^2 \tag{55}
\end{aligned}
$$

where the second inequality uses $\eta \le l^{-1}_{F,1}$, $\langle g_t, x_{t+1} - x_t \rangle \le -\frac{1}{\eta} \|x_{t+1} - x_t\|^2$ by [9, Lemma 3.1] and Cauchy-Schwartz inequality; the third applies Young's inequality

For simplicity, denote $h(x, y) = \gamma^{-1} f(x, y) + g(x, y)$, $v^h(x) = \min_{y \in \mathcal{Y}} h(x, y)$, $y^{\gamma,*}_t = \operatorname{arg min}_{y \in \mathcal{Y}} h(x_t, y)$ and $y^{g,*}_t \in \operatorname{arg min}_{y \in \mathcal{Y}} g(x_t, y)$, and the update bias $b(x_t) = \nabla F_\gamma(x_t) - g_t$. In this way,

$$
\begin{aligned}
\|b(x_t)\|^2 &= \|\nabla_x f(x_t, y^{\gamma,*}_t) + \gamma(\nabla_x g(x_t, y^{\gamma,*}_t) - \nabla_x g(x_t, y^{g,*}_t)) - \nabla_x f(x_t, y^\gamma_{t+1})\|^2 \\
&\overset{(a)}{\le} 2\|\nabla_x f(x_t, y^{\gamma,*}_t) - \nabla_x f(x_t, y^\gamma_{t+1})\| + 2\gamma^2 \|\nabla_x g(x_t, y^{\gamma,*}_t) - \nabla_x g(x_t, y^{g,*}_t)\|^2 \\
&\overset{(b)}{\le} 2l^2_{f,1} \|y_{t+1} - y^\gamma_t\|^2 + \mathcal{O}\big(\gamma^{-\frac{2(\alpha-1)}{2-\alpha}} + \delta \gamma\big)
\end{aligned}
$$

$$\overset{(c)}{\leq} \frac{4}{\mu_\gamma^*} l_{f,1}^2 (h(x_t, y_{t+1}^\gamma) - v^h(x_t)) + \mathcal{O}\big(\gamma^{-\frac{2(\alpha-1)}{2-\alpha}} + \delta\gamma\big)$$

$$\overset{(d)}{\leq} \frac{4}{\mu_\gamma^*} l_{f,1}^2 (1 - \eta^\gamma \mu)(h(x_t, y_t^\gamma) - v^h(x_t)) + \mathcal{O}\big(\gamma^{-\frac{2(\alpha-1)}{2-\alpha}} + \delta\gamma\big) \tag{56}$$

where (a) applies the Young's inequality; (b) follows the smoothness of $f$ and Lemma 1; (c) employs the property of strong convexity; and (d) is by the descent theory for applying projected gradient descent on problems satisfying PL condition, see e.g. [41, Theorem 5].

Plugging (56) in (55), there is

$$F_\gamma(x_{t+1}) - F_\gamma(x_t) \leq -\frac{\eta}{4} \|\nabla F_\gamma(x_t)\|^2 + \eta \frac{4}{\mu_\gamma^*} l_{f,1}^2 (1 - \eta^\gamma \mu)(h(x_t, y_t^\gamma) - v^h(x_t))$$

$$+ \eta \mathcal{O}\big(\gamma^{-\frac{2(\alpha-1)}{2-\alpha}} + \delta\gamma\big) \tag{57}$$

Moreover, as $h(x,y)$ is $l_{h,1} = \gamma^{-1} l_{f,1} + l_{g,1}$-smooth and $v^h(x)$ is $l_{v^h,1} = l_{h,1}(1 + L_y^\gamma)$-smooth, there is

$$h(x_{t+1}, y_{t+1}^\gamma) - v^h(x_{t+1})$$

$$\overset{(a)}{\leq} h(x_t, y_{t+1}^\gamma) - v^h(x_t) + \langle \nabla_x h(x_t, y_{t+1}^\gamma) - \nabla v^h(x_t), x_{t+1} - x_t \rangle + \frac{\eta^2(l_{h,1} + l_{v^h,1})}{2} \Big\| \frac{x_{t+1} - x_t}{\eta} \Big\|^2$$

$$\overset{(b)}{\leq} h(x_t, y_{t+1}^\gamma) - v^h(x_t) + \eta l_{h,1} \|y_{t+1}^\gamma - y_t^{\gamma,*}\| \Big\| \frac{x_{t+1} - x_t}{\eta} \Big\| + \frac{\eta^2(l_{h,1} + l_{v^h,1})}{2} \Big\| \frac{x_{t+1} - x_t}{\eta} \Big\|^2$$

$$\overset{(c)}{\leq} h(x_t, y_{t+1}^\gamma) - v^h(x_t) + \eta l_{h,1} \frac{z}{2} \|y_{t+1}^\gamma - y_t^{\gamma,*}\|^2 + \frac{\eta l_{h,1}}{2z} \Big\| \frac{x_{t+1} - x_t}{\eta} \Big\|^2 + \frac{\eta^2(l_{h,1} + l_{v^h,1})}{2} \Big\| \frac{x_{t+1} - x_t}{\eta} \Big\|^2$$

$$\overset{(d)}{\leq} (1 + \frac{\eta l_{h,1} z}{2})(h(x_t, y_{t+1}^\gamma) - v^h(x_t)) + (\frac{\eta l_{h,1}}{2z} + \frac{\eta^2(l_{h,1} + l_{v^h,1})}{2}) \Big\| \frac{x_{t+1} - x_t}{\eta} \Big\|^2$$

$$\overset{(e)}{\leq} (1 + \frac{\eta l_{h,1} z}{2})(1 - \eta^\gamma \mu)(h(x_t, y_t^\gamma) - v^h(x_t)) + (\frac{\eta l_{h,1}}{2z} + \frac{\eta^2(l_{h,1} + l_{v^h,1})}{2}) \Big\| \frac{x_{t+1} - x_t}{\eta} \Big\|^2, \quad \forall z > 0. \tag{58}$$

Here, (a) follows the smoothness of $h(x,y) + v^h(x)$ in $x$; (b) applies Cauchy-Schwartz inequality and the smoothness of $h$ in $y$; (c) uses Young's inequality for any $z > 0$; (d) is from the PL condition of $h(x,y)$ in $y$; (e) is similarly by the descent theory for applying projected gradient descent on $h(x, \cdot)$ satisfying PL condition [41, Theorem 5].

In this way, adding $c(h(x_{t+1}, y_{t+1}^\gamma) - v^h(x_{t+1}))$ to both side of (56), there is

$$F_\gamma(x_{t+1}) + c(h(x_{t+1}, y_{t+1}^\gamma) - v^h(x_{t+1}))$$

$$\leq F_\gamma(x_t) + \Big( -\frac{\eta}{4} + c\Big( \frac{\eta l_{h,1}}{2z} + \frac{\eta^2(l_{h,1} + l_{v^h,1})}{2} \Big) \Big) \Big\| \frac{x_{t+1} - x_t}{\eta} \Big\|^2$$

$$+ c\Big( \big(1 + \eta\big(\frac{l_{h,1} z}{2} + l_{f,1}^2 \frac{4}{\mu_\gamma^* c}\big)\big)(1 - \eta^\gamma \mu)(h(x_t, y_t^\gamma) - v^h(x_t)) \Big) + \eta \mathcal{O}\big(\gamma^{-\frac{2(\alpha-1)}{2-\alpha}} + \delta\gamma\big).$$

In this way, choose the following hyper-parameter,

$$\begin{cases} c = \mu^{-\frac{1}{2}} \\ z = 8 c l_{h,1} \\ \eta^\gamma \leq l_{h,1}^{-1} \\ \eta \leq \min\Big\{ \frac{1}{8c(l_{h,1} + l_{v^h,1})}, \frac{\eta^\gamma \mu/(1 - \eta^\gamma \mu)}{\frac{l_{h,1} z}{2} + \frac{4}{\mu c} l_{f,1}^2} \Big\} \end{cases} \tag{59}$$

i.e. $c = \mathcal{O}(1)$, $\eta = \mathcal{O}(1)$, there is

$$F_\gamma(x_{t+1}) - F_\gamma(x_t) + c(h(x_{t+1}, y_{t+1}^\gamma) - v^h(x_{t+1}))$$

---

**Algorithm 2** Fully-single-loop F$^2$SA [45] without momentum

---

1: **inputs:** initial points $x_0$; step size $\eta$, $\eta^g$, $\eta^\gamma$; counters $T$.
2: **for** $t = 0, 1, \ldots, T-1$ **do**
3: $\quad$ update $y_{t+1}^g = y_t^g - \eta^g \nabla_y g(x_t, y_t)$
4: $\quad$ update $y_{t+1}^\gamma = y_t^\gamma - \eta^\gamma \big(\nabla_y \gamma^{-1} f(x_t, y_t^\gamma) + \nabla_y g(x_t, y_t^\gamma)\big)$
5: $\quad$ update $x_{t+1} = x_t - \eta g_t$ where $g_t = \nabla_x f(x, y_t^\gamma) + \gamma(\nabla_x g(x, y_t^\gamma) - \nabla_x g(x, y_t^g))$.
6: **end for**
7: **outputs:** $(x_T, y_t^\gamma)$

---

$$\leq -\frac{\eta}{8}\|\nabla F_\gamma(x_t)\|^2 + c(h(x_t, y_t^\gamma) - v^h(x_t)) + \eta \mathcal{O}\big(\gamma^{-\frac{2(\alpha-1)}{2-\alpha}} + \delta\gamma\big).$$

Denote $D_1 = F_\gamma(x_0) - F_\gamma(x_T)$, $D_2 = (h(x_0, y_0^\gamma) - v^h(x_0)) - (h(x_T, y_T^\gamma) - v^h(x_T))$. Rearranging and telescoping gives

$$\frac{1}{T}\sum_{t=0}^{T-1}\|\nabla F_\gamma(x_t)\|^2 \leq \frac{8(D_1 + cD_2)}{\eta T} + \mathcal{O}\big(\gamma^{-\frac{2(\alpha-1)}{2-\alpha}} + \delta\gamma\big)$$

$$= \mathcal{O}(T^{-1} + \delta^{\frac{2(\alpha-1)}{\alpha}}) \tag{60}$$

where the last equality is achieved as $c = \mathcal{O}(1)$ and $\eta = \mathcal{O}(1)$ and by setting $\gamma = \mathcal{O}(\delta^{-\frac{2-\alpha}{\alpha}})$. This confirms that the trajectory $x_t$ stabilizes on average. Moreover, the hyper-parameter choices in (59) reformulate (58), which can be plugged in (56) to obtain

$$\frac{1}{T}\sum_{t=0}^{T-1}\|b_t\|^2 \leq \frac{4}{\mu_\gamma^*}l_{f,1}^2(1-\eta^\gamma\mu)\frac{1}{T}\sum_{t=0}^{T-1}(h(x_t, y_t^\gamma) - v^h(x_t)) + \mathcal{O}\big(\delta^{\frac{2(\alpha-1)}{\alpha}}\big) \leq \mathcal{O}\big(T^{-1} + \delta^{\frac{2(\alpha-1)}{\alpha}}\big). \tag{61}$$

where the last inequality follows by applying Abel's summation formula on series $\sum_{k=1}^{K} a_k b_k$ where $a_k = \mathcal{O}((1-\eta^\gamma\mu/2)^k)$ and $K^{-1}\sum_{k=0}^{K} b_k = \mathcal{O}(T^{-1} + \delta^{\frac{2(\alpha-1)}{\alpha}})$.

In this way, there is

$$\|\nabla F_\gamma(x_t)\|^2 = \left\|\frac{x_t - (x_t - \eta\nabla F_\gamma(x_t))}{\eta}\right\|^2$$

$$= \left\|\frac{x_t - (x_t - \eta g_t - \eta b_t)}{\eta}\right\|^2$$

$$\leq 2\left\|\frac{x_t - x_{t+1}}{\eta}\right\|^2 + 2\|b_t\|^2 \tag{62}$$

where inequality is by Young's inequality. Therefore,

$$\frac{1}{T}\sum_{t=0}^{T-1}\|\nabla F_\gamma(x_t)\|^2 \leq 2\frac{1}{T}\sum_{t=0}^{T-1}\left\|\frac{x_t - x_{t+1}}{\eta}\right\|^2 + 2\frac{1}{T}\sum_{t=0}^{T-1}\|b_t\|^2$$

$$\leq \mathcal{O}\big(T^{-1} + \delta^{\frac{2(\alpha-1)}{\alpha}}\big) \tag{63}$$

where the last inequality is obtained by plugging in (60) and (61). $\alpha \leq 1.5$ and rearranging.

### B.7 Additional discussion on fully-single-loop version of F$^2$SA [45]

Since momentum updates in F$^2$SA [45] introduce additional memory cost, in this section, we look into the fully-single-loop version of F$^2$SA [45] without momentum, i.e. at each iteration $t$, it updates:

$$y_{t+1}^g = y_t^g - \eta^g \nabla_y g(x_t, y_t), \quad \text{and} \tag{64}$$

$$y_{t+1}^\gamma = y_t^\gamma - \eta^\gamma \big(\nabla_y \gamma^{-1} f(x_t, y_t^\gamma) + \nabla_y g(x_t, y_t^\gamma)\big) \tag{65}$$

where $\eta^g \leq l_{g,1}^{-1}$ and $\eta^\gamma \leq (l_{g,1} + \gamma^{-1}l_{f,1})^{-1}$ are the step sizes. We summarize the algorithm in Algorithm 2 present the convergence results in the following theorem.

**Proposition 5.** *Suppose all assumptions in Proposition 2 hold. For iterations using the fully-single-loop version of Algorithm 2 with $\eta = \mathcal{O}(\gamma^{-1})$ gives*

$$\frac{1}{T}\sum_{t=0}^{T-1}\|\nabla F_\gamma(x_t)\|^2 = \mathcal{O}(\gamma^2 T^{-1}). \tag{66}$$

*Proof.* Denote the gradient approximate $g_t = \nabla_x f(x, y_{t+1}^\gamma) + \gamma\nabla_x g(x, y_{t+1}^\gamma) - \gamma\nabla_x g(x, y_{t+1}^g)$. According to smoothness, we have

$$
\begin{aligned}
F_\gamma(x_{t+1}) - F_\gamma(x_t) \leq & \langle\nabla F_\gamma(x_t) - g_t + g_t, x_{t+1} - x_t\rangle + \frac{l_{F,1}}{2}\|x_{t+1} - x_t\|^2 \\
\leq & -\frac{1}{\eta}\|x_{t+1} - x_t\|^2 + \frac{1}{2\eta}\|x_{t+1} - x_t\|^2 + \|\nabla F_\gamma(x_{t+1}) - g_t\|\|x_{t+1} - x_t\| \\
\leq & -\frac{1}{2\eta}\|x_{t+1} - x_t\|^2 + \frac{1}{4\eta}\|x_{t+1} - x_t\|^2 + \eta\|\nabla F_\gamma(x_{t+1}) - g_t\|^2 \\
= & -\frac{\eta}{4}\|\nabla F_\gamma(x_t)\|^2 + \eta\|\nabla F_\gamma(x_{t+1}) - g_t\|^2
\end{aligned}
\tag{67}
$$

where the second inequality uses $\eta \leq l_{F,1}^{-1}$, $\langle g_t, x_{t+1} - x_t\rangle \leq -\frac{1}{\eta}\|x_{t+1} - x_t\|^2$ by [9, Lemma 3.1] and Cauchy-Schwartz inequality; the third applies Young's inequality

Moreover, denote $h(x, y) = \gamma^{-1}f(x, y) + g(x, y)$, $v^h(x) = \min_{y\in\mathcal{Y}} h(x, y)$, $y_t^{\gamma,*} = \arg\min_{y\in\mathcal{Y}} h(x_t, y)$ and $y_t^{g,*} \in \arg\min_{y\in\mathcal{Y}} g(x_t, y)$, by triangle inequality and Young's inequality, there is

$$
\begin{aligned}
& \|\nabla F_\gamma(x_t) - g_t\|^2 \\
\leq & 2\gamma^2\|\nabla_x h(x_t, y_{t+1}^\gamma) - \nabla v^h(x_t)\|^2 + 2\gamma^2\|\nabla_x g(x_t, y_{t+1}^g) - \nabla v(x_t)\|^2 \\
= & 2\gamma^2 l_{h,1}^2\|y_{t+1}^\gamma - y_t^{\gamma,*}\|^2 + 2\gamma^2 l_{g,1}^2\|y_{t+1}^g - y_t^{g,*}\|^2 \\
\leq & 2\gamma^2 l_{h,1}^2\frac{2}{\gamma\mu_h}\gamma(h(x_t, y_{t+1}) - v^h(x_t)) + 2\gamma^2 l_{g,1}^2\frac{2}{\gamma\mu}\gamma(g(x_t, y_{t+1}) - v(x_t)) \\
\leq & 2\gamma^2 l_{h,1}^2\frac{2}{\gamma\mu_h}(1 - \eta^y\mu_h)\gamma(h(x_t, y_t) - v^h(x_t)) + 2\gamma^2 l_{g,1}^2\frac{2}{\gamma\mu}(1 - \eta^y\mu)\gamma(g(x_t, z_t) - v(x_t)) \\
\leq & 2\gamma^2 l_{h,1}^2\frac{2}{\gamma\mu_h}(1 - \eta^y\mu_h)\gamma(h(x_t, y_t) - v^h(x_t)) + 2\gamma^2 l_{g,1}^2\frac{2}{\gamma\mu}(1 - \eta^y\mu)\gamma(g(x_t, z_t) - v(x_t)) \\
\leq & 2\gamma^2 l_{h,1}^2\frac{1}{\gamma^2\mu_h^2}(1 - \eta^y\mu_h)\|\gamma\nabla_y h(x_t, y_t)\|^2 + 2\gamma^2 l_{g,1}^2\frac{1}{\gamma^2\mu^2}(1 - \eta^y\mu)\|\gamma\nabla_y g(x_t, z_t)\|^2
\end{aligned}
\tag{68}
$$

The second to last inequality follows PL condition and the last inequality is by the descent theory for applying projected gradient descent on problems satisfying PL condition, see e.g. [41, Theorem 5].

Plugging (68) in (67), there is

$$
\begin{aligned}
F_\gamma(x_{t+1}) - F_\gamma(x_t) \leq & -\frac{\eta}{4}\|\nabla F_\gamma(x_t)\|^2 + \eta\frac{4}{\mu_\gamma^*}\gamma^2(\gamma^{-1}l_{f,1} + l_{g,1})^2(1 - \eta^\gamma\mu)(h(x_t, y_t^\gamma) - v^h(x_t)) \\
& + \eta\frac{4}{\mu}\gamma^2(l_{g,1})^2(1 - \eta^g\mu)(h(x_t, y_t^g) - v^h(x_t)).
\end{aligned}
\tag{69}
$$

Moreover, as $h(x, y)$ is $l_{h,1} = \gamma^{-1}l_{f,1} + l_{g,1}$-smooth and $v^h(x)$ is $l_{v^h,1} = l_{h,1}(1 + L_y^\gamma)$-smooth, there is

$$
\begin{aligned}
& h(x_{t+1}, y_{t+1}^\gamma) - v^h(x_{t+1}) \\
\overset{(a)}{\leq} & h(x_t, y_{t+1}^\gamma) - v^h(x_t) + \langle\nabla_x h(x_t, y_{t+1}^\gamma) - \nabla v^h(x_t), x_{t+1} - x_t\rangle + \frac{\eta^2(l_{h,1} + l_{v^h,1})}{2}\|\nabla F_\gamma(x_t)\|^2 \\
\overset{(b)}{\leq} & h(x_t, y_{t+1}^\gamma) - v^h(x_t) + \eta l_{h,1}\|y_{t+1}^\gamma - y_t^{\gamma,*}\|\|\nabla F_\gamma(x_t)\| + \frac{\eta^2(l_{h,1} + l_{v^h,1})}{2}\|\nabla F_\gamma(x_t)\|^2 \\
\overset{(c)}{\leq} & h(x_t, y_{t+1}^\gamma) - v^h(x_t) + \eta l_{h,1}\frac{z}{2}\|y_{t+1}^\gamma - y_t^{\gamma,*}\|^2 + \frac{\eta l_{h,1}}{2z}\|\nabla F_\gamma(x_t)\|^2 + \frac{\eta^2(l_{h,1} + l_{v^h,1})}{2}\|\nabla F_\gamma(x_t)\|^2
\end{aligned}
$$

$$\overset{(d)}{\leq}(1+\frac{\eta l_{h,1}z}{2})(h(x_t,y_{t+1}^{\gamma})-v^h(x_t))+(\frac{\eta l_{h,1}}{2z}+\frac{\eta^2(l_{h,1}+l_{v^h,1})}{2})\|\nabla F_\gamma(x_t)\|^2$$

$$\overset{(e)}{\leq}(1+\frac{\eta l_{h,1}z}{2})(1-\eta^\gamma\mu)(h(x_t,y_t^\gamma)-v^h(x_t))+(\frac{\eta l_{h,1}}{2z}+\frac{\eta^2(l_{h,1}+l_{v^h,1})}{2})\|\nabla F_\gamma(x_t)\|^2, \quad \forall z>0. \tag{70}$$

Here, (a) follows the smoothness of $h(x,y)+v^h(x)$ in $x$; (b) applies Cauchy-Schwartz inequality and the smoothness of $h$ in $y$; (c) uses Young's inequality for any $z>0$; (d) is from the PL condition of $h(x,y)$ in $y$; (e) is similarly by the descent theory for applying projected gradient descent on $h(x,\cdot)$ satisfying PL condition [41, Theorem 5].

Following similar analysis, as $g(x,y)$ is $l_{g,1}$-smooth and $v(x)$ is $l_{v,1}=l_{g,1}(1+L_y^g)$-smooth, there is

$$g(x_{t+1},y_{t+1}^g)-v(x_{t+1})$$
$$\leq(1+\frac{\eta l_{g,1}z'}{2})(1-\eta^g\mu_{g*})(g(x_t,y_t^g)-v(x_t))+(\frac{\eta l_{g,1}}{2z'}+\frac{\eta^2(l_{g,1}+l_{v,1})}{2})\|\nabla F_\gamma(x_t)\|^2. \tag{71}$$

In this way, adding $c(h(x_{t+1},y_{t+1}^\gamma)-v^h(x_{t+1}))$ and $c'(g(x_{t+1},y_{t+1}^g)-v(x_{t+1}))$ to both side of (69), there is

$$F_\gamma(x_{t+1})-F_\gamma(x_t)+c(h(x_{t+1},y_{t+1}^\gamma)-v^h(x_{t+1}))+c'(g(x_{t+1},y_{t+1}^g)-v(x_{t+1}))$$
$$\leq\left(-\frac{\eta}{4}+c\Big(\frac{\eta l_{h,1}}{2z}+\frac{\eta^2(l_{h,1}+l_{v^h,1})}{2}\Big)+c'\Big(\frac{\eta l_{g,1}}{2z'}+\frac{\eta^2(l_{g,1}+l_{v,1})}{2}\Big)\right)\|\nabla F_\gamma(x_t)\|^2$$
$$+c\Big((1+\eta\big(\frac{l_{h,1}z}{2}+\gamma^2 l_{h,1}^2\frac{4}{\mu_\gamma^* c}\big))(1-\eta^\gamma\mu)(h(x_t,y_t^\gamma)-v^h(x_t))\Big)$$
$$+c'\Big((1+\eta\big(\frac{l_{g,1}z'}{2}+\gamma^2 l_{g,1}^2\frac{4}{\mu c'}\big))(1-\eta^g\mu_g)(g(x_t,y_t^g-v(x_t))\Big).$$

In this way, choose the following hyper-parameter,

$$\begin{cases} c=\gamma\mu^{-\frac{1}{2}} \\ c'=\gamma\mu^{-\frac{1}{2}} \\ z=16cl_{h,1} \\ z'=16c'l_{g,1} \\ \eta^g\leq l_{g,1}^{-1} \\ \eta^\gamma\leq l_{h,1}^{-1} \\ \eta\leq\min\left\{\frac{1}{16c(l_{h,1}+l_{v^h,1})},\frac{1}{16c'(l_{g,1}+l_{v,1})},\frac{\eta^\gamma\mu/(1-\eta^\gamma\mu)}{\frac{l_{h,1}z}{2}+\frac{4\gamma^2 l_{h,1}^2}{\mu c}},\frac{\eta^g\mu_g/(1-\eta^g\mu_g)}{\frac{l_{g,1}z'}{2}+\frac{4\gamma^2 l_{g,1}^2}{\mu_g c}}\right\} \end{cases} \tag{72}$$

i.e. $c,c'=\mathcal{O}(\gamma), \eta=\mathcal{O}(\gamma^{-1})$, there is

$$F_\gamma(x_{t+1})-F_\gamma(x_t)+c(h(x_{t+1},y_{t+1}^\gamma)-v^h(x_{t+1}))+c'(g(x_{t+1},y_{t+1}^g)-v(x_{t+1}))$$
$$\leq-\frac{\eta}{8}\|\nabla F_\gamma(x_t)\|^2+c(h(x_t,y_t^\gamma)-v^h(x_t))+c'l_{g,1}^2(g(x_t,y_t^g-v(x_t)).$$

Denote $D_1=F_\gamma(x_0)-F_\gamma(x_T)$, $D_2=(h(x_0,y_0^\gamma)-v^h(x_0))-(h(x_T,y_T^\gamma)-v^h(x_T))$, and $D_3=(g(x_0,y_0^g)-v(x_0))-(g(x_T,y_T^g)-v(x_T))$. Rearranging and telescoping gives

$$\frac{1}{T}\sum_{t=0}^{T-1}\|\nabla F_\gamma(x_t)\|^2\leq\frac{8\,(D_1+cD_2+c'D_3)}{\eta T}=\mathcal{O}(\gamma^2 T^{-1}) \tag{73}$$

where the last equality is because $c,c'=\mathcal{O}(\gamma)$, and $\eta=\mathcal{O}(\gamma^{-1})$. $\qquad\square$

The convergence of the fully-single-loop F$^2$SA without momentum is hindered by a larger $\gamma$, which regulates the UL violation rate. While the general case requires $\gamma=\mathcal{O}(\epsilon^{-0.5})$ as per Lemma 1. This shows that the fully-single-loop version of F$^2$SA, though computationally efficient, suffers from higher international cost.

## C  Additional Experimental Details

### C.1  Additional details for toy example in Figure 2

In this section, we provide details for the toy example of PEFT BLO problem in Figure 2. We consider a binary classification setting where the model parameters $\theta = (x, y)$ consist of the UL variable $x$ and LL variable $y$, with $\theta \in \mathbb{R}^2$. The model implements a 1D convolutional network with softmax activation:

```
class SoftmaxNN(nn.Module):
    def __init__(self):
        super().__init__()
        self.hidden = nn.Conv1d(in_channels=1, out_channels=1,
                       kernel_size=2, stride=2, bias=False)
        self.activation = nn.Softmax(dim=1)
        self._init_weight()
```

We specify the SFT datasets $\mathcal{D}_{\text{SFT}} = \{(X_1, y)\}$, and the DPO dataset $\mathcal{D}_{\text{DPO}} = \{(X_2, y_w, y_\ell)\}$ in Table 3. The BLO problem is specified in (3), where $f_{\text{DPO}}$ consists of a DPO loss with $\beta = 1$ [70] plus an $\ell_2$ regularization term (weight 0.01) and $g_{\text{SFT}}$ consists of a negative log-likelihood loss and the same regulariztion. The reference model is obtained via learning on $g_{\text{SFT}}(x, y)$ (parameterized with $(x = -5.34, y = -9.94)$).

We apply our PBGD-Free algorithm in Algorithm 1 in comparison with F$^2$SA [45] with $\gamma = 15$, $K = 10$ inner loop to solve (5), and $T = 5000$ outer loop for both algorithms.

Table 3: Dataset specification for toy illustration

| Input | Output | Feature |
|:-----:|:------:|:-------:|
| $X_1$ | $y = 0$ | $[1.0, 1.0, 0.5, 0.5]^\top$ |
| $X_1$ | $y' = 1$ | $[1.0, 0.5, 0, 0.5]^\top$ |
| $X_2$ | $y_w = 1$ | $[1.0, 0.5, 0.5, 0.5]^\top$ |
| $X_2$ | $y_\ell = 0$ | $[0.5, 1.0, 1.0, 1.0]^\top$ |

### C.2  Representation learning problem on NLSY dataset [73]

BLO has proven effective in representation learning for obtaining a joint backbone model $x$ that captures unified task features and generalizes well to downstream tasks by only tuning the head $y$ [4, 95, 83, 36, 80]. We test our algorithm on a representation learning problem on the National Longitudinal Survey of Youth (NLSY) dataset [73], following the experimental setup in [80]. This problem aims to learn representations to predict normalized income via $\min_{x,y} f_{\text{MSE}}(x, y; D_1)$ s.t. $y \in \arg\min_y f_{\text{MSE}}(x, y; D_2)$, where $D_1, D_2$ are datasets partitioned by gender. The representation model, parameterized $x$, consists of two fully connected layers (hidden size 200, ReLU activation), and the predictor, parameterized $y$, is a linear classification head.

We compare our fully-single-loop PBGD-Free (with inner iteration $K = 1$) against F$^2$SA [45] (with $K = 2$) and the ITD algorithm from [80], following the experimental setup in [80]. As shown in Table 4, the performance gap is particularly notable in efficiency, where PBGD-Free is over twice as fast as F$^2$SA [45] and more than 30 times faster than the ITD-based approach [80], primarily because it omits the value-function part and the inner loop of $y_g^*(x)$. Moreover, PBGD-Free achieves lower MSE than PBGD [44]. This improvement stems from PBGD-Free's ability to avoid the bias $\gamma$-propagation inherent in PBGD's design. When both algorithms are single-loop (or nearly single-loop for small $K = 2$), PBGD's reliance on a fixed penalty parameter $\gamma$ amplifies initial inner update biases throughout training, slowing convergence, detailed in Appendix B.7 while PBGD-Free eliminates these $\gamma$-dependent value function terms.

### C.3  LLM PEFT problem (3)

**General Setup.**  We evaluate our PEFT framework (3) using the `Dahoas/rm-hh-rlhf` dataset for DPO loss and the `OpenOrca` dataset for SFT loss. For training, we test one PYTHIA-1b [6]

| Methods | MSE | Time (s) |
|---|---|---|
| F$^2$SA [45] | $1.9331 \pm 0.0794$ | $12.33 \pm 0.34$ |
| Implicit [80] | $2.1530 \pm 0.0455$ | $169.69 \pm 0.36$ |
| **PBGD-Free** | $\mathbf{1.8916 \pm 0.1245}$ | $\mathbf{5.15 \pm 0.06}$ |

Table 4: Performance results for different training methods on representation learning problem on NLSY-7k Dataset [73]. The mean $\pm$ standard deviation is reported for both the mean MSE and the mean time over 5 random experiments on the test dataset.

| Method | Avg Memory Used (MB) | Peak Memory Used (MB) |
|---|---|---|
| BOME | 18834.53 | 21535.96 |
| F$^2$SA | 16213.78 | 17622.43 |
| ALRIGHT | 16031.86 | **16107.45** |
| **PBGD-Free** | **16016.94** | 16180.89 |

Table 5: Empirical GPU memory usage for each method over 3 epoches of training.

model with $1800$ samples for each dataset (batch size 16) and the LLAMA-3-3B [22] model with $4800$ samples (batch size 32). Both models are adapted with LoRA (ALPHA 16, RANK 16) and we treat LoRA PEFT weights on the attention layers as $x$, the last layer linear head as $y$. The learning rate is set to $1 \times 10^{-5}$, using Adam [43] as the optimizer. All experiments were conducted on a cluster of NVIDIA A6000 GPUs, each with 40 GB of memory. Training was performed using PyTorch with the DeepSpeed library https://github.com/deepspeedai/DeepSpeed to optimize memory usage and distributed training efficiency. We consider a time-limited experiment under a consistent computational budget, reflecting real-world constraints where training time is often a critical factor.

**Algorithm hyperparameter.** We use a penalty constant of $\gamma = 10$ for our proposed PBGD-Free algorithm (Algorithm 1) with a single inner loop ($K = 1$). For the baseline F$^2$SA algorithm [11, 45], we set $\gamma = 10$ with $K = 3$ inner updates for training LLAMA-3-3B [31], and $K = 5$ for PYTHIA-1b [6]. For the BOME algorithm, we similarly use $K = 3$ and $K = 5$ inner loops, adopting its hyperparameter $\eta = 0.5$ for calculating the penalty constant, as suggested in [105]. For the ALRIGHT algorithm [24], we use its default setting of $\lambda = 0.5$ as suggested in literature [24]. Since the ALRIGHT algorithm in [24] is a bi-objective learning algorithm that does not have the representation learning capability, we examine it on an alternative formulation $\min_{x,y}[f_{\text{DPO}(x,y)}, g_{\text{SFT}}(y)]$.

**Faster training than BLO baselines and more stable over bi-objective.** As presented in Figure 9, when training PYTHIA-1b [6] on the PEFT problem (3), our PBGD-Free algorithm demonstrates the fastest convergence compared to the baseline BLO methods F$^2$SA [11, 45] and BOME [105], both of which fail to converge within the given time budget. Additionally, PBGD-Free shows greater stability compared to its bi-objective counterpart ALRIGHT [24]. Table 5 reports the average and peak GPU memory consumption over 3 epochs for all compared methods, illustrating that PBGD-Free maintains a memory footprint comparable to the baselines.

**Better transferability to new task.** Figure 10 further illustrates the performance of the outputs from BLO PEFT learning (3) in the subsequent post-SFT-tuning phase **S2)**. The BLO baselines (F$^2$SA [11, 45] and BOME [105]), which did not achieve convergence in the initial PEFT phase due to their higher time complexity, tend to sacrifice DPO performance when improving SFT performance during post-SFT tuning **S2)**. In contrast, PBGD-Free algorithm and its bi-objective counterpart ALRIGHT [24] demonstrate the ability to preserve strong preference alignment (DPO) while conducting SFT training in **S2)**. This shows that the preference backbone $x$ learned by both of them can be adapted to new task by fine-tuning only the linear head to achieve strong SFT performance. Notably, the BLO PEFT outputs trained by PBGD-Free achieves better SFT performance with substantially lower SFT loss, highlighting the advantage of the prioritization of SFT in our BLO formulation (3). This structure allows for a more powerful SFT tuning head, whereas bi-objective training methods tend to oscillate between potentially conflicting objectives, thereby limiting their post-SFT performance.

## Example of SFT Evaluation Performance

**Human**: Generate an approximately fifteen-word sentence that describes all this data: Midsummer House eatType restaurant; Midsummer House food Chinese; Midsummer House priceRange moderate; Midsummer House customer rating 3 out of 5; Midsummer House near All Bar One

| **PYTHIA-1b [6]**: | **BLO-PEFT (F²SA [45])**: | **BLO-PEFT (PBGD-Free)**: |
|---|---|---|
| Midsummer House staff a restaurant priced restaurant restaurant. a good-5 star rating. and in All Bar One. | Midsummer House is a restaurant priced restaurant restaurant with a 3-5 customer rating. located near All Bar One. | Midsummer House is a moderately priced Chinese restaurant with a 3/5 customer rating. located near All Bar One. |

**Human**: You will be given a definition of a task first, then some input of the task. This task is about using the specified sentence and converting the sentence to Resource Description Framework (RDF) triplets of the form (subject, predicate, object). The RDF triplets generated must be such that the triplets accurately capture the structure and semantics of the input sentence. The input is a sentence and the output is a list of triplets of the form [subject, predicate, object] that capture the relationships present in the sentence. When a sentence has more than 1 RDF triplet possible, the output must contain all of them. AFC Ajax (amateurs)'s ground is Sportpark De Toekomst where Ajax Youth Academy also play.

| **LLAMA-3-3B [31]**: | **BLO-PEFT (F²SA [45])**: | **BLO-PEFT (PBGD-Free)**: |
|---|---|---|
| [["AjaxFC Ajax (amateurs)", "playsGround", "Sportpark De Toekomst"], ["Ajax Youth Academy", "has at", "Sportpark De Toekomst"]] | [["AjaxFC Ajax (amateurs)", "plays ground", "Sportpark De Toekomst"], ["Ajax Youth Academy", "has at", "Sportpark De Toekomst"]] | [["AFC Ajax (amateurs)", "has ground", "Sportpark De Toekomst"], ["Ajax Youth Academy", "plays at", "Sportpark De Toekomst"]] |

Table 6: Examples of SFT evaluation performance for PYTHIA-1b [6], LLAMA-3-3B [31] and their corresponding BLO-PEFT (3) results via our PBGD-Free Algorithm 1 and basline F²SA [45]. Text marked in red indicates incorrect outputs, orange indicates partially correct outputs that follow some of the instructions, and green indicates fully correct outputs that match the expected instructions.

**Better SFT performance while maintaining preference learning.** As illustrated in Table 6, the outputs generated by PBGD-Free demonstrate more precise and semantically accurate extraction, highlighting its superior SFT performance. In Figure 12, we present the loss metrics performance throughout post-SFT-tuning phase for LLAMA-3-3B [31] in addition to the results presented in Section 4. We observe that our backbone $x$ trained on BLO PEFT (3) via PBGD-Free retains its lowest DPO rates throughout post-SFT-tuning. Figure 11 shows the quantitative results of the preference alignment using average reward gap and win rate. Together with Figure 8, they indicate that the backbone preference model $x$ by the PBGD-Free maintains the first-tier DPO performance for preference alignment while enhancing the SFT performance by only fine-tuning the linear head. The slight DPO drop in Figure 11 of PBGD-Free compared with ALRIGHT is because it prioritizes better SFT performance, which restricts the feasible search space of representation model optimizing at the UL. However, since the representation evaluation criterion prioritizes strong SFT performance achieved by fine-tuning only the linear head, and treats preference alignment as a secondary goal, PBGD-Free remains the top-performing method. Moreover, according to Figure 7, PBGD-Free is more stable during the training compared with ALRIGHT. Additionally, Table 6 provides the SFT output comparison given by PBGD-Free and F²SA on PYTHIA-1b [6], LLAMA-3-3B [31], from which we can see that both methods improve the response quality over the pre-trained model through BLO PEFT (3), while PBGD-Free generates better responses and follows the human instructions well.

**Higher-rank LoRA enables finding better preference backbone via PBGD-Free.** The last 2 columns in Figure 6 show that a higher-rank LoRA better preserves DPO with comparable SFT performance. It is likely because a higher-rank LoRA provides more over-parameterization, which ensures a more benign optimization landscape for the representation parameter $x$ [101, 88, 54] and thus enables globally finding a better representation model $x$ [90].

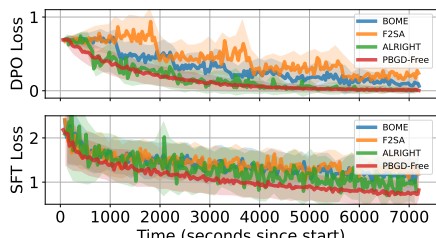

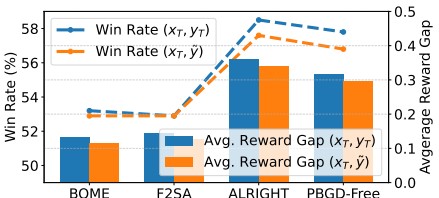

Figure 9: Train losses vs. time with STRIDE = 50 for different algorithms in solving (3) (or biobjective learning for ALRIGHT [24]) on PYTHIA-1b [6].

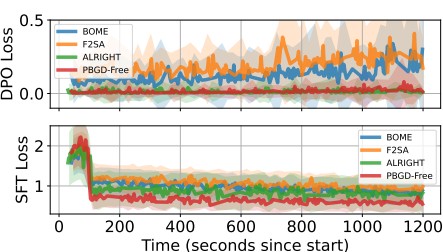

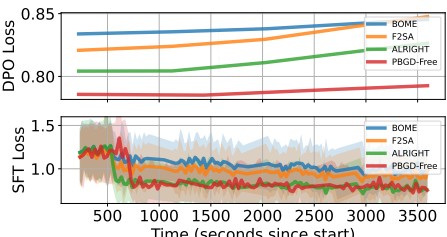

Figure 10: Train losses vs. time with STRIDE = 50 for different algorithms in Post SFT-tuning phase on PYTHIA-1b [6].

Figure 11: Average Reward Gap ($\uparrow$) and Win Rate ($\uparrow$) for different algorithms for PEFT LLAMMA-3-3B on [31] with the output $(x_T, y_T)$ via each method in **S1)** and the outcome $(x_T, \tilde{y})$ from post-SFT-tuning on another dataset with fixed-backbone in **S2)**.

Figure 12: Evaluation of DPO losses and train SFT loss vs. time with STRIDE = 50 for different algorithms in Post SFT-tuning phase on for PEFT LLAMMA-3-3B [31].

## C.4 BiDoRa fine-tuning problem

One of the recent applications of bilevel optimization lies in the field of Large Language Finetuning. [68] proposed BiDoRa, which considers fine-tuning using DoRa [53] by training on a BLO problem

$$\min_m l^l_{\text{tr}}(m, v^*(m)) \quad \text{s.t} \quad v^*(m) = l^s_{\text{tr}}(m, v^*(m) + \rho R(v)) \tag{74}$$

where $m$ is the magnitude and $v$ is the direction matrix for the low-rank incremental direction, $l^l_{\text{tr}}(m, v)$ and $l^l_{\text{tr}}(m, v)$ are respectively loss functions for fine-tuning training dataset splitting into large and small on proportion 0.66 to 0.67, and $R(v)$ is the Gram regularization loss [94] with constant $\rho$ taken as $1e^{-3}$.

We conduct experiments on Microsoft Research Paraphrase Corpus (MRPC) dataset [20], and Internet Movie Database (IMDb) in Hugging Face by fine-tuning Bert model [59]. We apply fully-single-loop versions of PBGD and PBGD-Free in Algorithm 1 to solve the problem in Section 74 and compare it with training using DARTS [48], the algorithm used in the original BiDoRa algorithm [68], and the naive results trained on $\min_{m,v} l_{tr}(m, v)$ where $l_{tr}$ is the combined loss for training dataset including the ones used for both $l^l_{tr}$ and $l^s_{tr}$ for DoRa [53]. The experiment is conducted on a single NVIDIA RTX A5000 GPU (24GB) using CUDA 12.2 and NVIDIA driver version 535.113.01.

As illustrated in Table 7, training BiDoRa using PBGD-Free in Algorithm 1 achieves the best performance in terms of test accuracy. It is more efficient than training using PBGD as it cuts the inner loop. Notably, it performs even better than the fully-single-loop of F$^2$SA [11]. This is consistent with the convergence results in Proposition 5 and Theorem 3.

| Methods | MRPC | IMDb |
|---|---|---|
| **BiDoRa-PBGD-Free** | **0.839 $\pm$ 0.006** | **0.873 $\pm$ 0.007** |
| BiDoRa-F$^2$SA | 0.820 $\pm$ 0.014 | 0.866 $\pm$ 0.016 |
| BiDoRa-DARTS | / | / |
| DoRa | 0.832 $\pm$ 0.010 | 0.872 $\pm$ 0.010 |

Table 7: Test accuracy (%) on training the finetuning parameters using BiDoRa-PBGD in comparison with DARTS [48], the algorithm used in [68], and with directly training DoRa [53]. It represents the accuracy mean $\pm$ standard deviation on 20 random training experiments. The "/" represents didn't converge in 10 times the time used in training DoRa.

## D Broader Impact

This paper mainly focuses on developing an efficient BLO algorithm with a theoretical guarantee. By applying the proposed method to representation learning based PEFT and BiDORA, our work contributes to the broader development of parameter efficient LLM fine-tuning. Potential societal impacts include applications in creative content generation, data augmentation, and machine learning-based simulations. While we acknowledge the possibility of unintended uses, we do not identify any specific societal risks that need to be highlighted in this context.

