# OpenReview forum: "Beyond Value Functions: Single-Loop Bilevel Optimization under Flatness Conditions"
_NeurIPS.cc/2025/Conference — NeurIPS 2025 poster_

### Official Review · Reviewer_z3Zu · 2025-06-25

**Clarity:** 2
**Significance:** 2
**Originality:** 3
**Rating:** 4
**Confidence:** 4

**Summary:**

This paper proposes PBGD-Free fully single-loop algorithm for bilevel optimization problems that avoids costly value-function computations.  PBGD-Free simplifies the process by removing the need to solve the inner optimization problem at every iteration. The authors show that while the method may fail to converge under standard Lipschitz assumptions, introducing a relaxed flatness condition—modeled by a H\o lder-like inequality—ensures convergence to near-optimal solutions. They theoretically establish that under this flatness condition, PBGD-Free achieves favorable iteration complexity and low memory costs. Extensive experiments on parameter-efficient fine-tuning (PEFT) of LLMs demonstrate that PBGD-Free matches or outperforms state-of-the-art baselines in both accuracy and efficiency, validating the proposed theoretical framework.

**Questions:**

1. Line 38, it would be clearer if the authors explained what the smoothness constant refers to. Do they mean the Lipschitz constant of the gradient?
2. Line 42 $F$ should be $\tilde F$?

**Ethical Concerns:**

["NO or VERY MINOR ethics concerns only"]

**Final Justification:**

My key concern about the delta is addressed.

**Limitations:**

yes.

**Paper Formatting Concerns:**

no.

**Quality:**

2

**Strengths And Weaknesses:**

Strength:
1. This work proposes a relaxed assumption for Lipschitz continuity and provides a convergence analysis.
2. The method empirically performs well.

Weakness:
1.  The paper is not very clearly written. DPO and its notations are not clearly introduced. For example, it would be clearer if the meanings of $x$ and $y$ in the model were explicitly specified. Additionally, it is unclear what $\sigma$ represents in equation (13), and how $\nabla f_{\text{DPO}}$ is derived.
2. The assumption on $\delta$ in the flatness condition is strong. It requires $\delta$ to be smaller than $\epsilon$ to allow for a smaller choice of $\lambda$. More importantly, Theorem 3 requires $\delta(x_t)$ to be summable over $t$. If $T$ goes to infinity, then $\delta(x_T)$ must tend to zero.
4.  The complexity analysis still reveals that the proposed method may not converge. In the main result, Theorem 3, the proposed method includes an extra term in the complexity bound that depends on $\delta$. Therefore, although the experiments show promising empirical results, the method lacks theoretical convergence guarantees, which makes it less satisfactory.

---

> ### Author Rebuttal · Authors · 2025-07-31
>
> We thank the reviewer for the thoughtful evaluation and for recognizing our contributions, including the relaxed assumption and strong empirical results. Some concerns appear to stem from misinterpretations, and we appreciate the opportunity to clarify these points. Our responses follow below.
>
> **W1. DPO notation not clearly introduced: meaning of x,y in (3); what $\sigma$ is in eq.(13); how $\nabla f_{DPO}$ is derived**
>
> The decomposition of LLM model parameter $(x, y)$ in Eq.(3) is introduced in Section 3.1 and further contextualized in Section 4.1. Specifically, $x$ denotes the parameters of the pretrained LLM backbone, while $y$ corresponds to a lightweight, fine-tunable head (e.g., a linear layer). This type of decomposition is standard in LLM literature [Pfeiffer et al., 2020] [Zaken et al., 2021]. While it is discussed in the text, we agree that presenting it more explicitly earlier when introducing Eq.(3) would aid readability. We will revise the manuscript accordingly.
>
> In Eq.(13), $\sigma(\cdot)$ refers to the sigmoid function, commonly used in DPO-based losses [66]. The gradient $\nabla f_{\text{DPO}}$ is derived in a standard manner by differentiating the DPO loss with respect to the model parameters $(x, y)$ according to [66], leveraging the softmax-based parameterization
> $$
> \pi_{x,y}(r | z) := \text{softmax}(\text{model}_{x,y}(z))_r.
> $$
>
> We follow the same conventions and setup as in the original DPO formulation in [66].
>
> We appreciate the reviewer’s suggestions and will revise the manuscript to clarify these notations earlier in the paper to improve readability.
>
> Reference
> >- Pfeiffer, Jonas, Aishwarya Kamath, Andreas Rücklé, Kyunghyun Cho, and Iryna Gurevych. "Adapterfusion: Non-destructive task composition for transfer learning." arXiv preprint arXiv:2005.00247 (2020).
> >- Zaken, Elad Ben, Shauli Ravfogel, and Yoav Goldberg. "Bitfit: Simple parameter-efficient fine-tuning for transformer-based masked language-models." arXiv preprint arXiv:2106.10199 (2021).
>
>
> **W2. The flatness condition is strong. $\delta < \epsilon$ required for small $\lambda$. Theorem 3 requires $\delta(x_t)$ to be summable over $t$.**
>
> We thank the reviewer for raising this important point. Since we did not mention “small $\lambda$” in our paper as suggested by the reviewer, we assume the reviewer is referring to the requirement for small $\delta$ in our convergence analysis.
>
> Indeed, if $\delta$ is not sufficiently small, convergence may only occur within a neighborhood of the stationary point. However,
>
> > **Flatness with small $\delta$ is NOT strong. Usually $\delta \approx 0$ in real-world deep learning problems (our focus).**
>
> We support this claim both theoretically (in Observations 1–3) and empirically (PEFT example in Section 3.1, 4; additional experiments in Appendix C). We elaborate further here.
>
> To recap, the flatness parameter $\delta$ defined in Definition 1 is a generalized upper bound from a standard Hölder-type inequality:
>
> $|f(x, y^*(x)) - f(x, y')| \leq c \|y' - y^{\star}(x)\|^\alpha + \delta(x), \forall y.$
>
> When $\delta = 0$, classical Hölder theory \[Evans, 1993] implies that this inequality can only hold for $\alpha \leq 1$ or for $\nabla_y f(x, y) = 0$. Thus, to accommodate higher-order decay $\alpha > 1$, which is crucial for convergence analysis, we relax to allow $\delta > 0$.
>
> To further elaborate, consider the behavior of $\delta(x)$ under different local structures of $f(x, y)$. Here, we consider local strucuture as $y_\gamma$ is controled by $\gamma$ to be close to $y_g$.
>
> * **Negative result on locally linear** case: If $f(x, y) \approx f(x, y^{\star}(x)) + \langle \nabla_y f(x, y^{\star}(x)), y - y^{\star}(x) \rangle$, then $\delta(x) := \max_{y'} \| \nabla_y f(x, y^{\star}(x))\|\|y' - y^{\star}(x)\| - c \|y' - y^{\star}(x)\| ^\alpha = \max_{z\geq 0}  \| \nabla_y f(x, y^{\star}(x))\| z- c z^\alpha=  \frac{\alpha - 1}{\alpha} \alpha^{-\frac{1}{\alpha -1}} \| \nabla_y f(x, y^{\star}(x))\|$, choosing $c =\| \nabla_y f(x, y^{\star}(x))\|$.
> So although $\delta(x)$ can be artitrarily small when $\alpha\rightarrow 1$ even if $\| \nabla_y f(x, y^{\star}(x))\|$ is not small. When we plug this into eq.(17), the residual will be $\| \nabla_y f(x, y^{\star}(x))\|^{\frac{2(\alpha-1)}{\alpha}} (\frac{1}{\alpha})^{\frac{2}{\alpha}}(1-\frac{1}{\alpha})^{2(1-\frac{1}{\alpha})}\leq \| \nabla_y f(x, y^{\star}(x))\|^{\frac{2(\alpha-1)}{\alpha}}$, which yields constant error. This is consistent with our negative Example 1, but is still tighter than the ${\cal O}(\| \nabla_y f(x, y^{\star}(x))\|^2)$ bound for large $\| \nabla_y f(x, y^{\star}(x))\|$ under traditional Lipschitzness Assumption 1.
>
> * **Our focus on locally curved** case: However, for most of **deep learning problem**, using losses e.g. cross entropy, $\ell_2$ loss, exponential, etc, are hardly linear: $f(x, y) \approx f(x, y^{\star}(x)) + {\cal O}(\|y - y^{\star}(x)\|^\alpha), \quad \alpha > 1$, then $\delta(x)= {\cal O}(\|y - y^{\star}(x)\|^\alpha) - c\|y - y^{\star}(x)\|^\alpha$ **naturally becomes 0**.
>
> **Practical deep learning loss landscapes are highly non-linear and rarely locally linear in the parameter space.** Our PEFT example in Section 3.1, 4 and Observations 1–3 show that the flatness condition with **small $\delta(x)$ is consistently satisfied** across a variety of tasks and models.
>
>
> Regarding Theorem 3, it might be a **misunderstanding**. In fact, we **DO NOT** require $\delta(x_t)$ to be summable. Rather, we only require the **average** $\frac{1}{T} \sum_{t=0}^{T-1} \delta(x_t)$ to be small, which does not necessarily imply $\delta(x_T)\rightarrow 0$ when $T$ goes to infinity. For example, consider a sequence where $\delta(x_t) = 1$ when $t = 2^n$ for integer $n$, and $\delta(x_t) = 1/t$ otherwise. This sequence is **not summable**, and $\delta(x_T)$ **does not** converge to zero. However, the average remains small because the majority of $\delta(x_t)$ values are small. This is a rather mild condition and is aligned with practical use.
>
>
> We would also like to emphasize the **fundamental difficulty** of the bilevel optimization setting considered in our paper. Earlier work either require double loop [40, 44, 45, 50, 73, 74, 100] or suffers higher computational complexity [15, 28, 33, 39, 42, 47, 78] (or both). Designing efficient **first-order** and **fully single-loop algorithm** with convergence guarantees is an open challenge. To the best of our knowledge, our work is the **first to address this challenge**, via proposing a mild and verifiable flatness condition.
>
>
> We respectfully invite the reviewer to **reconsider** the contribution in light of this clarification. We will revise the manuscript to make this point clearer, highlighting both the theoretical intuition and the empirical justification in realistic deep learning contexts. Thank you again for highlighting this subtle and important issue.
>
>
> **W3. The complexity analysis shows the algorithm may not converge.**
>
> This might be a **misunderstanding**. The algorithm can converge depending on the flatness of the objective function. Please refer to the answer for **W2**.
>
> **Q1 (minor). Explain the smoothness constant in line 38.**
>
> We appreciate the reviewer’s attention to detail. Yes, the “smoothness constant” refers to the Lipschitz constant of the gradient of the corresponding objective function. We will clarify this terminology in the revision.
>
> **Q2 (minor). Typo: $F$ should be $\tilde{F}$**
>
> Thank you for catching this typo. We confirm that it should be $\tilde{F}$, and we will correct it in the revised version.
>
>
> We hope these clarifications help resolve the concerns and offer a clearer view of our contributions. We would be grateful if the reviewer could take our responses into account when reassessing the paper and consider the possibility of raising the score.

---

### Official Review · Reviewer_gkxV · 2025-06-25

**Clarity:** 2
**Significance:** 3
**Originality:** 3
**Rating:** 4
**Confidence:** 4

**Summary:**

Due to the nested problem structure, most existing algorithms require either the Hessian vector calculation or the nested loop updates, which are computationally inefficient in large language model (LLM) fine-tuning. Building upon the fully first-order penalty-based approach, this paper proposes an efficient value function-free (PBGD-Free) algorithm that eliminates the loop of solving the lower-level problem and admits fully single-loop updates.

**Questions:**

1. In Table 1, what do $d_x$ and $d_y$ represent?

2. Has Definition 1 been previously used in the literature? If so, relevant citations should be provided.

3. Observation 2 should explicitly state the smoothness conditions imposed on $f$.

4. In Lemma 3, the notation $x^*$ always denotes the optimal solution, which may cause ambiguity. A distinct notation should be introduced. Moreover, Additionally, the range $\alpha\in (1,1.5]$ appears restrictive—does this hold for all functions, or could $\alpha = 2$ be feasible?


5. The algorithm should be reformulated as a single-loop structure, eliminating the dependence on $K$.

6. In line 279, what is $l_F^{-1}$?


7. In Equation (17), can $\delta$ be made arbitrarily small? If not, the algorithm’s output accuracy would remain fixed regardless of $T$, which is undesirable.


If the author can provide sufficient reasons or evidence to prove the effectiveness of the proposed algorithm and the adopted assumptions, I'm willing to increase my score.

**Ethical Concerns:**

["NO or VERY MINOR ethics concerns only"]

**Final Justification:**

The author's response addressed all of my concerns, and I have no further questions.

**Limitations:**

See **Weaknesses** and **Questions**.

**Quality:**

2

**Strengths And Weaknesses:**

**Strengths**: 1. This paper proposes PBGD-Free, a computationally efficient fully-single-loop, value-function-free, first-order algorithm.

2. This paper introduces a Holder-alike condition to describe the flatness of $f$, which relaxes the standard $l_{f,0}$-Lipschitz continuity assumption when $l_{f,0}$ is small.

3. This paper is well-written.

4. The figures in this paper look very good, for example, Figure 2.


**Weaknesses**: 1. Assumption 2 imposes a Lipschitz condition on $\nabla^2 f$, which appears overly restrictive. Standard requirements in the literature typically do not necessitate such an assumption.

2. Furthermore, the $\mu$-PL condition imposed on $c f(x,y) + g(x,y)$ in Assumption 2 seems excessively strong, as the PL condition for the sum of functions is rarely encountered in prior work. Additionally, the dynamic nature of $c$ further complicates the satisfaction of this condition. While the authors argue that the PL condition is weaker than strong convexity, they should provide more concrete examples to substantiate this claim, particularly when $g$ lacks strong convexity.

3. The format of references is not consistent.

---

> ### Author Rebuttal · Authors · 2025-07-31
>
> We thank the reviewer for recognizing our contribution and presentation. Our response to your comments follows.
>
> **W1-2. Assumption on: Lipschitz condition on hessians of $f$ and $g$; PL condition (Assumption 2) for the sum of functions $cf+g$ for small $c$. How PL is weaker than SC. examples?**
>
> We thank the reviewer for raising these important points. We explained the feasibility of the assumptions in Line 147-150. All asumptions in Assumption 2 are traditional in Bilevel literature  [11, 15, 19, 28, 33, 38, 45, 74].
>
> The Lipschitz condition on Hessians of $f$ and $g$ is standard in literature [45, 11, 19] and used to establish that the smooth constant $l_{F,1}$ of $F_\gamma$ is ${\cal O}(1)$ independent from $\gamma$. This condition enables us to choose step sizes and to prove convergence rates. Even if this assumption is relaxed, convergence still holds under appropriately chosen step sizes $\eta = {\cal O}(1/\gamma)$.
>
> PL condition is strictly weaker than assuming strong convexity. Specifically, all strongly convex functions satisfy the PL inequality, the converse is not true. For example, $f(x,y) = \frac{1}{2} (x - 2 \sin(y))^2$ is known to satisfy the PL condition but is not strongly convex, illustrating this gap.
>
> Moreover, assuming that $c f + g$ satisfies PL for sufficiently small $c$ is a commonly adopted assumption in bilevel literature [45, 74, 11]. This assumption is crucial for the existence of the nested gradient $\nabla \phi(x)$. This assumption is weaker than strongly convexity. For example, $g(y)=(y^2-1)^2$ is non-convex but PL, $f(y)=-y^2$ is non-convex, and $cf(y)+g(y)$ is 4-PL for all $c\leq 0.25$.
>
> **W3. (Minor) Format of references is not consistent.**
>
> Thank you for pointing this out. We will correct the reference formatting to ensure consistency in the revised version.
>
> **Q1. $d_x$ and $d_y$ in table 1**
>
> $d_x$ and $d_y$ are dimension for variable $x$ and $y$. It is firstly introduced in Line 19. We will make it more explicit in the revision.
>
> **Q2. Is Def 1 in previous literature?**
>
> **No, Definition 1 is novel and introduced in this work.** We designed it specifically to support our goal of developing the first efficient, first-order, single-loop bilevel algorithm with convergence guarantees. This definition captures a mild (see observation 1-3) and practically verifiable condition (see the calculation rule in (12)). It holds in many real-world settings, such as the PEFT scenario discussed in Section 3.1.
>
> Importantly, Definition 1 plays a key role in enabling our theoretical guarantees and serves as a foundation for what we believe is a promising first step toward solving bilevel problems efficiently in this regime. Additionally, the assumption on flatness condition with small $\delta$ is mild for deep-learning problems, as we later elaborated in response to your Q7.
>
> We appreciate the opportunity to clarify this contribution and will highlight it in the revised version.
>
> **Q3. (Minor) Observation 2 should explicitly state the smoothness conditions imposed on $f$**
> We thank the reviewer for pointing this out. We will correct this to "Under the $l_{f,1}$-smoothness condition of $f(x,\cdot)$" to make it more clear.
>
> **Q4. In Lemma 3, $x^*$ may cause ambiguity; $\alpha \in (1,1.5]$ may be strong.**
>
> We appreciate the reviewer’s careful reading. Regarding the notation $x^*$, we intended it to denote a stationary point rather than a global optimum. While this usage is common in optimization literature, we acknowledge that it may cause ambiguity. We will revise the notation or clarify its meaning explicitly in the text to avoid confusion.
>
> Regarding the range $\alpha \in (1, 1.5]$, this condition arises from the flatness assumption on $f(x^*, \cdot)$, and reflects a trade-off between the regularity of the function near the lower-level optimum and the convergence guarantees needed in bilevel optimization. Specifically, larger $\alpha$ and smaller $\delta$ imply flatter behavior around the optimum, which leads to a more restrictive conditions on $f$.
>
> Notably, the ideal case of $\delta = 0$ (pure Hölder continuity) can only be achieved if $\alpha \leq 1$ or $\nabla_y f(x, y_g^*(x)) =0$, as dictated by the textbook with Hölder continuity theory [Section 5.1; Evans, 1993]. To accommodate $\alpha > 1$, we introduce a relaxed flatness condition in Definition 1 by allowing $\delta > 0$. In this formulation, a larger $\delta$ (e.g., $\delta = 2$) reflects a tighter or more restrictive requirement on the behavior of $f$, while the range $\alpha \in (1, 1.5]$ remains moderate. In this way, although larger $\alpha$ is feasible, it is not needed.
>
> Reference
> >- Evans, Lawrence C. Partial differential equations. Vol. 19. American mathematical society, 1993.
>
> **Q5. The algorithm is single-loop, and the dependence of $K$ should be removed.**
>
> We agree that in both our implementation and experiments, we use $K = 1$, and this is sufficient for the setting considered in this work. We have annotated this choice in Algorithm 1 to make it clear.
>
> We chose to retain the inner-loop presentation because it facilitates the derivation of our theoretical results, particularly the negative result in Proposition 2. This structure allows for a direct comparison with classical bilevel approaches based on value function gradients, and helps highlight the contrast between our method and such approximations.
>
> We will revise the presentation slightly to clarify that $K = 1$ is the default in our setting.
>
> **Q6. What is $l_{F}^{-1}$ in line 276?**
>
> The notation $l_{F,1}$ is the smoothness constant of $F_\gamma(x)$. It is not scalable with $\gamma$, i.e. ${\cal O}(1)$, according to [11]. We will clarify this.
>
> **Q7. Can $\delta$ in eq.(17) be arbitrarily small?**
>
> We appreciate the reviewer’s thoughtful question. In our convergence analysis, the parameter $\delta$ captures the degree of flatness of the upper-level objective around the lower-level solution $y^*(x)$, and it directly influences how closely our algorithm converges to a true stationary point. Indeed, if $\delta$ is not sufficiently small, convergence may only occur within a neighborhood of the stationary point.
>
> To directly answer the reviewer’s question:
> > **$\delta$ is generally very small (usually 0) in real-world deep learning problems.**
>
> We support this claim both theoretically (in Observations 1–3) and empirically (PEFT example in Section 3.1, 4; more experiments in Appendix C). We elaborate further here.
>
> To recap, the flatness parameter $\delta$ defined in Definition 1 is a generalized upper bound from a standard Hölder-type inequality:
>
> $|f(x, y^*(x)) - f(x, y')| \leq c \|y' - y^{\star}(x)\|^\alpha + \delta(x), \forall y.$
>
> When $\delta = 0$, classical Hölder theory \[Evans, 1993] implies that this inequality can only hold for $\alpha \leq 1$ or for $\nabla_y f(x, y) = 0$. Thus, to accommodate higher-order decay $\alpha > 1$, which is crucial for convergence analysis, we relax to allow $\delta > 0$.
>
> To further elaborate, consider the behavior of $\delta(x)$ under different local structures of $f(x, y)$. Here, we consider local strucuture as $y_\gamma$ is controled by $\gamma$ to be close to $y_g$.
>
> * **Negative result on locally linear** case: If $f(x, y) \approx f(x, y^{\star}(x)) + \langle \nabla_y f(x, y^{\star}(x)), y - y^{\star}(x) \rangle$, then $\delta(x) := \max_{y'} \| \nabla_y f(x, y^{\star}(x))\|\|y' - y^{\star}(x)\| - c \|y' - y^{\star}(x)\| ^\alpha = \max_{z\geq 0}  \| \nabla_y f(x, y^{\star}(x))\| z- c z^\alpha=  \frac{\alpha - 1}{\alpha} \alpha^{-\frac{1}{\alpha -1}} \| \nabla_y f(x, y^{\star}(x))\|$, choosing $c =\| \nabla_y f(x, y^{\star}(x))\|$.
>
> So although $\delta(x)$ can be artitrarily small when $\alpha\rightarrow 1$ even if $\| \nabla_y f(x, y^{\star}(x))\|$ is not small. When we plug this into eq.(17), the residual will be $\| \nabla_y f(x, y^{\star}(x))\|^{\frac{2(\alpha-1)}{\alpha}} (\frac{1}{\alpha})^{\frac{2}{\alpha}}(1-\frac{1}{\alpha})^{2(1-\frac{1}{\alpha})}\leq \| \nabla_y f(x, y^{\star}(x))\|^{\frac{2(\alpha-1)}{\alpha}}$, which yields constant error. This is consistent with our negative Example 1, but is still tighter than the ${\cal O}(\| \nabla_y f(x, y^{\star}(x))\|^2)$ bound for large $\| \nabla_y f(x, y^{\star}(x))\|$ under traditional Lipschitzness Assumption 1.
>
> * **Our focus on locally curved** case: However, for most of **deep learning problem**, using losses e.g. cross entropy, $\ell_2$ loss, exponential, etc, are hardly linear: $f(x, y) \approx f(x, y^{\star}(x)) + {\cal O}(\|y - y^{\star}(x)\|^\alpha), \quad \alpha > 1$, then $\delta(x)= {\cal O}(\|y - y^{\star}(x)\|^\alpha) - c\|y - y^{\star}(x)\|^\alpha$ **naturally becomes 0**.
>
> **Practical deep learning loss landscapes are highly non-linear and rarely locally linear in the parameter space.** Our PEFT example in Section 3.1, 4 and Observations 1–3 show that the flatness condition with **small $\delta(x)$ is consistently satisfied** across a variety of tasks and models.
>
> We would also like to emphasize the **fundamental difficulty** of the bilevel optimization setting considered in our paper. Earlier work either require double loop [40, 44, 45, 50, 73, 74, 100] or suffers higher computational complexity [15, 28, 33, 39, 42, 47, 78] (or both). Designing efficient **first-order** and **fully single-loop algorithm** with convergence guarantees is an open challenge. To the best of our knowledge, our work is the **first to address this challenge**, via proposing a mild and verifiable flatness condition.
> We respectfully invite the reviewer to **reconsider** the contribution in light of this clarification. We will revise the manuscript to make this point clearer, highlighting both the theoretical intuition and the empirical justification in realistic deep learning contexts. Thank you again for highlighting this subtle and important issue.

---

> > ### Comment · Reviewer_gkxV · 2025-08-01
> >
> > The author's response addressed all of my concerns, and I have no further questions. I will raise the rating to 4.

---

> > > ### Author Response · Authors · 2025-08-01
> > >
> > > Thanks so much for your thoughtful review and for letting us know that your questions were addressed. We really appreciate your support and are glad to hear you are raising the score.

---

### Official Review · Reviewer_txG5 · 2025-06-29

**Clarity:** 2
**Significance:** 2
**Originality:** 3
**Rating:** 4
**Confidence:** 4

**Summary:**

The paper proposes a method called PBGD-Free for bilevel optimization under a flatness condition. Compared with prior methods such as F$^2$SA that needs to solve two lower-level problems of the same condition number, the PBGD-Free only solves one, which make it more efficient in practice. Another benefit of the flatness condition is that it allows using single-loop updates,  while prior works without this condition requires $O(\log \epsilon^{-1})$ number of inner loops.

**Questions:**

See the weakness part.

**Ethical Concerns:**

["NO or VERY MINOR ethics concerns only"]

**Final Justification:**

I have raised the score to 4.

**Limitations:**

See the weakness part.

**Paper Formatting Concerns:**

I do not notice any major formatting issues in this paper.

**Quality:**

3

**Strengths And Weaknesses:**

**Strength** Each claim in the paper is well-supported by examples, theorems and experiments. The paper introduces a novel method that is more efficient in practice and provide conditions when the method can converge.

**Weakness**

1. The story of the article is self-consistent, but it doesn't completely convince me. For example, maybe I can come up with another simple explanation: in practice especially neural network training experiments, the network is over-parameterized such that $g^\ast(x)= 0 $. Consequently, the penalty function $ f(x,y) + \gamma (g(x,y) - g^\ast(x))$ is equivalent to  $f(x,y) + \gamma (g(x,y) $. Therefore, you can also skip the inner loop to approximate $g^\ast(x)$. I think my story is also self-consistent and can be supported by experiments.

2. I can not understand the necessity of using bilevel optimization in LLM PEFT experiments. Why could the bilevel PEFT be better than the standard PEFT training? I think it is still very uncommon to use bilevel optimization in PEFT.

3. The parameter setting of $\gamma = \Omega(\delta^{-\frac{2-\alpha}{\alpha }}) $ in Line 810 is not correct. For instance, if $\delta=0$ then such a setting indicates that $\gamma = +\infty$. I think the parameters from Line 810 to 817 should be checked, although I believe the final result is still correct.

4. (Minor) The reference [36] by Feihu Huang et al., is flawed, and I think it should be removed from the citations.

---

> ### Author Rebuttal · Authors · 2025-07-31
>
> We thank the reviewer for recognizing that "each claim in the paper is well-supported by examples, theorems and experiments", and for providing insightful suggestions. Our response for your comments follows:
>
> **W1. Whether the problem can be explained via $f+\gamma g$ while $v(x)=\min_y g(x,y)=0$?**
>
> We thank the reviewer for this insightful observation. Indeed, in certain **over-parameterized settings**, where the lower-level (LL) objective $g(x, y)$ can be minimized to zero for any $x$, the bilevel problem reduces to optimizing $f(x, y) + \gamma g(x, y)$.
>
> However, this condition does **not generally hold** in practical scenarios such as **PEFT**. In PEFT, the LL variable $y$ typically represents a **lightweight linear head**, rather than a highly over-parameterized module. As a result, $\min_y g(x, y)$ will not achieve zero (or constant).
>
> In fact, **the flatness assumption can still hold even when the LL problem is not over-parameterized**. This subtle but important point is emphasized and formally discussed in Section 3.1 of our paper.
> To further understand this, we conducted a follow-up experiment based on the PEFT setup described in Appendix C.1. This example is non-overparameterized with only 2 variables, and directly minimizing $f + \gamma g$ led to a solution that was more than 5 units away (in Euclidean distance) from the bilevel solution. Although we are unable to include plots in this rebuttal, we hope this observation helps illustrate the necessity of preserving the bilevel structure in non-overparameterized cases like PEFT. However, this example features flatness, as discussed in Section 3.1, while our PBGD-Free algorithm leads to convergence to bilevel solution.
>
> Moreover, directly omitting the value function $v(x)$ when it is zero leads to a different algorithm than ours. PBGD-Free updates $x$ using $\nabla\_{x} f(x, y^{\star}\_{\gamma} (x))$,
> whereas the latter uses $\nabla\_{x} f(x, y^{\star}\_{\gamma} (x))$ + $\gamma \nabla\_{x} g(x, y^{\star}\_{\gamma} (x))$.
> For analyzing the convergence of PBGD-Free, $v(x) = 0$ is insufficient, as discarding the  $\gamma \nabla\_{x} g(x, y^{\star}\_{\gamma} (x))$ term introduces an ${\cal O}(1)$ error.
>
> **W2. Why using Bilevel in LLM PEFT and what is its benefit over standard PEFT?**
>
> We thank the reviewer for this insightful question and the opportunity to clarify the motivation and the benefit behind our bilevel optimization framework for parameter-efficient fine-tuning (PEFT), introduced in Eq. (3):
>
> - Upper Level: DPO objective, optimizing backbone parameters $x$;
> - Lower Level: SFT objective, optimizing a lightweight linear head $y$.
>
> Our design is motivated by both theoretical considerations and practical needs in LLM deployment, as briefly discussed in Section 4.1 and Appendix C.3. Due to space constraints, we did not expand on these points in the main text, but we re-clarify them here:
>
> (1) It operates in a post-SFT setting, where DPO is fine-tuned from a pretrained model and SFT is applied to downstream tasks. This setup reflects the standard practice in LLM deployment, where models are adapted rather than trained from scratch, and human preferences are often shared across downstream tasks.
>
> (2) The bilevel structure is to mitigate catastrophic forgetting commonly observed in traditional PEFT pipelines. Sequential fine-tuning, where first applying SFT and then DPO—DPO updates, can inadvertently overwrite task-specific knowledge acquired during SFT [24]. Our bilevel formulation mitigates this by optimizing the DPO objective (UL) while conditioning on a near-optimal SFT solution (LL). This hierarchical structure helps preserve downstream task performance by maintaining task-specific adaptations in the head, while guiding backbone updates toward better preference alignment.
>
> (3) In real-world applications (e.g., industry use cases), it is common to fine-tune only a lightweight head for specific tasks while keeping the backbone fixed or lightly tuned [Pfeiffer et al., 2020] [Zaken et al., 2021]. Our method aligns with this PEFT practice by allowing the head to specialize for SFT tasks, while learning the preference backbone with DPO for generalization and alignment.
>
> We appreciate the reviewer’s suggestion and will revise the manuscript to explicitly highlight these three motivations for using the bilevel framework in PEFT.
>
> Reference
> >- Pfeiffer, Jonas, Aishwarya Kamath, Andreas Rücklé, Kyunghyun Cho, and Iryna Gurevych. "Adapterfusion: Non-destructive task composition for transfer learning." arXiv preprint arXiv:2005.00247 (2020).
> >- Zaken, Elad Ben, Shauli Ravfogel, and Yoav Goldberg. "Bitfit: Simple parameter-efficient fine-tuning for transformer-based masked language-models." arXiv preprint arXiv:2106.10199 (2021).
>
> **W3. The parameter setting of $\gamma={\cal O}(\delta^{-\frac{2-\alpha}{\alpha}})$ in Line 810. If $\delta = 0$ such indicates $\gamma = \infty$.**
>
> We thank the reviewer for the detailed review and insightful comment. In practice, we do not need to take $\delta = 0$, as we only aim to achieve a target accuracy $\epsilon > 0$. Specifically, to guarantee $\epsilon$-stationarity, we only need to choose $\delta^{\frac{2(\alpha - 1)}{\alpha}} = \epsilon$, i.e. select $\delta = \epsilon^{\frac{\alpha}{2(\alpha - 1)}}$. This leads to $\gamma = {\cal O}(\delta^{-\frac{2 - \alpha}{\alpha}}) = {\cal O}(\epsilon^{-\frac{2 - \alpha}{2(\alpha - 1)}})$.
>
> This approach aligns with standard non-asymptotic analyses in optimization, where parameters are chosen to ensure convergence to an $\epsilon$-stationary point for some small (but usually nonzero) $\epsilon$. We acknowledge that our original presentation may have been unclear, and we will revise the text to explicitly reflect this dependency and avoid potential misunderstandings.
>
> **W4. (Minor) The reference [36] by Huang et al., is flawed, and should be removed.**
>
> We thank the reviewer for pointing that out. We will remove it.
>
> We sincerely thank the reviewer for acknowledging the strengths of our empirical results and for the thoughtful questions that allowed us to clarify key theoretical points. We hope our detailed responses help resolve the remaining concerns, and we would be grateful if the reviewer could take them into consideration when finalizing their evaluation and possibly consider a higher score.

---

> ### Comment · Reviewer_txG5 · 2025-08-03
>
> I thank the authors for their detailed response. I do not have further questions. I think the condition that allows designing single-loop penalty methods is interesting.  I will raise the score to 4.

---

> > ### Author Response · Authors · 2025-08-05
> >
> > We thank the reviewer for acknowledging our contribution and for considering raising score. Please let us know if you have further questions.

---

### Official Review · Reviewer_8EAT · 2025-07-03

**Clarity:** 3
**Significance:** 3
**Originality:** 3
**Rating:** 4
**Confidence:** 4

**Summary:**

This paper addresses the computational inefficiency of bilevel optimization (BLO) algorithms often used in machine learning applications such as fine-tuning large language models (LLMs). Traditional BLO methods typically require expensive Hessian-vector computations or nested loop updates, which are impractical for large-scale problems. To overcome these challenges, the authors propose a computationally efficient, fully single-loop algorithm called PBGD-Free that eliminates the need for lower-level value function evaluations while retaining first-order updates. By leveraging a new flatness (Hölder-alike) condition for the upper-level objective, they provide theoretical guarantees of convergence for their method. Empirical results on parameter-efficient LLM fine-tuning and other bilevel tasks demonstrate that PBGD-Free achieves significant improvements in efficiency while matching or surpassing the accuracy of previous state-of-the-art methods.

**Questions:**

1. The authors state that the proposed method is more computationally and memory efficient than existing methods. However, no empirical evidence is provided to substantiate this claim. I recommend including results such as (1) convergence versus wall-clock time, and (2) peak memory usage for the proposed method and the baselines.

2. In LLM applications, DPO is typically applied as a post-SFT stage. Employing bi-level optimization in this context is uncommon. Additionally, MT-Bench is often used as a standard testbed under this setting instead of objective value or BLUE scores.

3. Adversarial attack scenarios or mitigating catastrophic forgetting are more common considerations in this area. I strongly encourage the authors to explore and evaluate their approach in these settings.

**Ethical Concerns:**

["NO or VERY MINOR ethics concerns only"]

**Limitations:**

Yes

**Quality:**

3

**Strengths And Weaknesses:**

**Strengths**:

1. This paper presents a novel, single-loop, value-function-free algorithm for bi-level optimization. The authors provide theoretical guarantees of convergence, and the proposed method empirically outperforms existing approaches.

2. The introduction of a new flatness condition to establish the convergence of the algorithm is notable. This condition may also prove valuable for future research in this area.

**Weaknesses**:

1. The paper does not discuss the empirical computational and memory costs in comparison to existing methods. (Please see my comments in Question 1.)

2. The experimental problem formulation does not reflect a typical setting for LLMs. (Please see my comments in Questions 2 and 3.)

---

> ### Author Rebuttal · Authors · 2025-07-31
>
> We thank the reviewer for appreciating our work and the constructive review. Our response to your comments follows.
>
> **W1 & Q1. Empirical computational and memory costs. (1) convergence versus wall-clock time, (2) peak memory usage for the proposed method and the baselines.**
>
> | Method        | Avg Memory Used (MB) | Peak Memory Used (MB) |
> | ------------- | -------------------- | --------------------- |
> | BOME      | 18834.53             | 21535.96              |
> | F²SA      | 16213.78             | 17622.43              |
> | ALRIGHT   | 16031.86             | 16107.45              |
> | **PBGD-Free** | **16016.94**     | **16180.89**          |
>
> We sincerely thank the reviewer for this insightful suggestion.
>
> For **wall-clock time versus convergence**, we have reported these results in Figures 7 and 9 in Appendix for the PEFT experiments. Our method, PBGD-Free, consistently demonstrates faster convergence and achieves superior final performance compared to the baselines. Additionally, Table 4 presents a summary of efficiency on the NLSY representation learning task, where PBGD-Free attains the best performance with the shortest wall-clock time. We appreciate the reviewer’s attention to this aspect and look forward to further clarifying these results in the revision.
>
> For **memory costs**, we provide empirical memory usage measurements during the rebuttal period for completeness. The table above summarizes the average and peak GPU memory consumption over 3 epochs for each method. PBGD-Free exhibits a memory footprint comparable to the baselines, underscoring its practical efficiency. Coupled with its superior convergence and performance, this highlights the strengths of our approach. We are grateful to the reviewer for encouraging us to share these valuable insights.
>
>
> **W2. Motivation and design of experimental setting for LLM PEFT.**
>
> We thank the reviewer for this insightful question, and the opportunity to clarify the motivation behind our bilevel optimization framework for parameter-efficient fine-tuning (PEFT), introduced in Eq. (3):
>
> - Upper Level: DPO objective, optimizing backbone parameters $x$;
> - Lower Level: SFT objective, optimizing a lightweight linear head $y$.
>
> Our design is motivated by both theoretical considerations and practical needs in LLM deployment, as briefly discussed in Section 4.1 and Appendix C.3. Due to space constraints, we did not expand on these points in the main text, but we re-clarify them here:
>
> (1) It operates in a post-SFT setting, where DPO is fine-tuned from a pretrained model and SFT is applied to downstream tasks. This setup reflects standard practice in LLM deployment, where models are adapted rather than trained from scratch, and human preferences are often shared across downstream tasks.
>
> (2) Bilevel structure is to mitigate the catastrophic forgetting issue in sequentially fine-tuning SFT followed by DPO [24], as DPO may overwrite task-specific knowledge. The hierarchical structure in the bilevel formulation enables optimizing DPO conditioned on a near-optimally learned SFT, helping to preserve downstream task performance.
>
>
> (3) In real-world applications (e.g., industry use cases), it is common to fine-tune only a lightweight head for specific tasks while keeping the backbone fixed or lightly tuned [Pfeiffer et al., 2020] [Zaken et al., 2021]. Our method aligns with this PEFT practice by allowing the head to specialize for SFT tasks, while learning the preference backbone with DPO for generalization and alignment.
>
>
> We appreciate the reviewer’s suggestion and will revise the manuscript to explicitly highlight these three motivations for using the bilevel framework in PEFT.
>
> Reference
> >- Pfeiffer, Jonas, Aishwarya Kamath, Andreas Rücklé, Kyunghyun Cho, and Iryna Gurevych. "Adapterfusion: Non-destructive task composition for transfer learning." arXiv preprint arXiv:2005.00247 (2020).
> >- Zaken, Elad Ben, Shauli Ravfogel, and Yoav Goldberg. "Bitfit: Simple parameter-efficient fine-tuning for transformer-based masked language-models." arXiv preprint arXiv:2106.10199 (2021).
>
>
> **Q2. MT-Bench is often used as a standard testbed under this setting instead of objective value or BLEU scores.**
>
> We appreciate the suggestion to use MT-Bench for evaluating alignment. However, we would like to clarify that our use of BLEU is solely for assessing supervised fine-tuning (SFT) quality, not alignment. The BLEU score measures how closely the model’s output matches the reference by evaluating the overlap of words and short sequences (n-grams), providing a standard metric for generation quality. It is a well accepted metrics for SFT performance [Papineni et al., 2002]. BLEU scores and training loss illustrate that our bilevel optimization effectively improves language generation under supervised objectives.
>
> For alignment, we report the Average Reward Gap (Appendix Fig. 11), computed from a learned reward model. This directly reflects the policy's alignment with implicit preferences. Together with the loss objectives $f_{DPO}$, $g_{SFT}$ , these metrics show that our bilevel framework learns a DPO-aligned model conditioned on near-optimal SFT. Specifically, it shows the best supervised performance across baselines (see BLEU score and $g_{SFT}$ in Table 2, Figure 8), and great DPO alignment.
>
> Due to **the lack of OpenAI API access**, we are unable to run MT-Bench evaluations, which rely on GPT-4-based judgments. We believe our current evaluation setup captures both supervised and alignment perspectives, and we plan to incorporate MT-Bench in future work when resources allow.
>
> **Q3. Consider Catastrophic forgetting and adversarial learning.**
> We thank the reviewer for the insightful suggestions. In fact, one of our motivations for adopting a bilevel framework is to address challenge of catastrophic forgetting, which is effectively mitigated by the hierarchical structure of the bilevel formulation. See details in Point (2) to W2.
>
> The focus of this paper is to tackle the technical challenge of solving bilevel optimization problems using a fully single-loop, first-order algorithm — a problem that is notoriously difficult. Our goal is to develop a method with efficient convergence behavior that matches standard gradient descent, while being scalable and practical. To illustrate this, we use LLM fine-tuning as a concrete and motivating application.
>
> We agree that both catastrophic forgetting and adversarial robustness are compelling application areas for our method and look forward to exploring them in future work.
>
> We thank the reviewer again for recognizing the contribution and novelty of our work. We hope our responses help address any remaining concerns and further highlight the strength of our results. We would be grateful if the reviewer could take these clarifications into account during the final evaluation and consider a higher score.

---

> > ### Comment · Reviewer_8EAT · 2025-08-04
> >
> > The authors' response has partially addressed my concerns. However, most of the suggested experiments have not been included. Therefore, I am maintaining my original score.

---

> > > ### Author Response · Authors · 2025-08-04
> > >
> > > We thank the reviewer for the thoughtful feedback. We would like to respectfully clarify that we have included additional empirical results and analysis in response to the review comments:
> > >
> > > 1. **Memory Cost Experiments:** We conducted new experiments to measure both average and peak GPU memory usage over 3 epochs. As reported in the rebuttal, our method (PBGD-Free) achieves the lowest memory footprint among bilevel optimizer baselines, demonstrating its practical efficiency.
> > >
> > > 2. **Clarification of Evaluation Metrics:** While we did not use MT-Bench due to **lack of access to the OpenAI API**, we clarified that BLEU scores are used solely to assess supervised fine-tuning (SFT) quality. For alignment evaluation, we report the **Average Reward Gap** computed from a learned reward model. Together, these two metrics capture both generation quality and alignment effectiveness. We acknowledge that MT-Bench is a valuable benchmark for alignment and plan to include it in the camera-ready version if resources allow.
> > >
> > > 3. **Catastrophic Forgetting:** We explicitly addressed catastrophic forgetting as a key motivation for adopting a bilevel framework (see Point (2) under W2 in the rebuttal). Additionally, **Appendix C.3** discusses the improved **transferability to new tasks**, providing empirical evidence that our method effectively preserves SFT knowledge — indicating that catastrophic forgetting is mitigated.
> > >
> > > 4. **Adversarial Learning:** While adversarial robustness is a compelling direction, it falls outside the scope of our current **theory-focused** work. We view it as a promising application area and hope to explore it in future research.
> > >
> > > We hope this clarifies that we took the reviewer’s suggestions seriously and responded with new experimental data and extended discussions. We sincerely appreciate the reviewer’s insights and remain committed to further improving our work.

---

### Note · Authors · 2025-08-15

We thank all reviewers for their constructive feedback and engagement during the discussion. All reviewers confirmed they would raise/keep their scores to 4 after the rebuttal. Below, we summarize our key contributions and how we address the reviewers' concerns.

**Key Contributions**

1. We propose **PBGD-Free** (Alg. 1), the **first** practical **first-order, fully single-loop** algorithm for **nonconvex–nonconvex bilevel optimization** with ${\cal O}(\epsilon^{-1})$ convergence guarantees—matching nonconvex gradient descent—**without any inner-loop computation** (Thm. 3).
2. We introduce a **novel flatness condition** (Def. 1), inspired by a relaxed Hölder continuity, that is **mild, verifiable, and naturally satisfied** in deep learning settings such as **PEFT** in LLMs. It requires only the **average flatness** over iterations to be small, broadening applicability (Thm. 3).
3. We validate our assumptions and method through **theoretical analysis, motivating examples, and real-world experiments** (Obs. 1–3; Sec. 4; Appx. C), showing **lowest memory usage** (Table 1; rebuttal to Reviewer 8EAT) and competitive **wall-clock convergence** (Figs 7, 9; Table 4).

**Resolution of Concerns**
We clarified on the **theoretical contributions**: the novelty and mildness of the flatness assumption, its theoretical underpinnings, and empirical validity; showed why simpler single-level reductions fail in realistic PEFT scenarios; and confirmed that our LL-PL assumption is standard in BLO literature. On the **empirical aspect**, we added new efficiency experiments, and clarified our PEFT design to reflect realistic deployment while mitigating catastrophic forgetting.

**Closing**
Our work makes a **significant advance** in BLO by providing the first efficient, single-loop algorithm with convergence guarantees under realistic assumptions, validated on large-scale deep learning tasks. All reviewer concerns have been addressed, and the contribution directly tackles a long-standing open challenge, offering clear value to the NeurIPS community.

---

### Decision · Program_Chairs · 2025-09-17

**Decision:**

Accept (poster)

**Comment:**

The authors define a new average flatness condition and, under this condition, provide an algorithm, with a convengence guarantee, for bilevel non-convex optimization. The reviewers found strength in the results; the condition was natural and the problem very important (e.g. useful for llm optimization). Technical details were discussed at length, but no major outstanding concerns remained. Some concerns about implementability of the algorithm were resolved, and the authors were convinging in their arguments of practicality of the algorithm. Considering that the proposed method has been motivated in a few independent ways -- theoretically and practically -- I think it will be a vaulable addition to the community.